# Metabolic reprogramming and altered cell envelope characteristics in a pentose phosphate pathway mutant increases MRSA resistance to β-lactam antibiotics

**Merve S. Zeden** [1]*, **Laura A. Gallagher**[1], **Emilio Bueno**[2], **Aaron C. Nolan**[1], **Jongsam Ahn**[3], **Dhananjay Shinde**[3], **Fareha Razvi**[3], **Margaret Sladek**[3], **Órla Burke**[1], **Eoghan O'Neill**[4], **Paul D. Fey**[3], **Felipe Cava**[2], **Vinai C. Thomas**[3], **James P. O'Gara** [1]*

**1** Microbiology, School of Biological and Chemical Sciences, University of Galway, Galway, Ireland, **2** Department of Molecular Biology, Umeå University, MIMS—Laboratory for Molecular Infection Medicine Sweden, Umeå, Sweden, **3** Department of Pathology and Microbiology, University of Nebraska Medical Center, Omaha, Nebraska, United States of America, **4** Department of Clinical Microbiology, Royal College of Surgeons in Ireland, Dublin, Ireland

* merve.zeden@universityofgalway.ie (MSZ); jamesp.ogara@universityofgalway.ie (JPO'G)

**Data Availability Statement:** Whole-genome sequence data is available from the European Nucleotide Archive (https://www.ebi.ac.uk/ena),

## Abstract

Central metabolic pathways control virulence and antibiotic resistance, and constitute potential targets for antibacterial drugs. In *Staphylococcus aureus* the role of the pentose phosphate pathway (PPP) remains largely unexplored. Mutation of the 6-phosphoglucono-lactonase gene *pgl*, which encodes the only non-essential enzyme in the oxidative phase of the PPP, significantly increased MRSA resistance to β-lactam antibiotics, particularly in chemically defined media with physiologically-relevant concentrations of glucose, and reduced oxacillin (OX)-induced lysis. Expression of the methicillin-resistance penicillin binding protein 2a and peptidoglycan architecture were unaffected. Carbon tracing and metabolomics revealed extensive metabolic reprogramming in the *pgl* mutant including increased flux to glycolysis, the TCA cycle, and several cell envelope precursors, which was consistent with increased β-lactam resistance. Morphologically, *pgl* mutant cells were smaller than wild-type with a thicker cell wall and ruffled surface when grown in OX. The *pgl* mutation reduced resistance to Congo Red, sulfamethoxazole and oxidative stress, and increased resistance to targocil, fosfomycin and vancomycin. Levels of lipoteichoic acids (LTAs) were significantly reduced in *pgl*, which may limit cell lysis, while the surface charge of *pgl* cells was significantly more positive. A *vraG* mutation in *pgl* reversed the increased OX resistance phenotype, and partially restored wild-type surface charge, but not LTA levels. Mutations in *vraF* or *graRS* from the VraFG/GraRS complex that regulates DltABCD-mediated D-alanylation of teichoic acids (which in turn controls β-lactam resistance and surface charge), also restored wild-type OX susceptibility. Collectively these data show that reduced levels of LTAs and OX-induced lysis combined with a VraFG/GraRS-dependent increase in cell surface positive charge are accompanied by significantly increased OX resistance in an MRSA *pgl* mutant.

Registered Project PRJEB59981, sample accession numbers ERS14733509-ERS14733512 and run accession numbers ERR10960504-ERR10960507. The SAUSA300_FRP3757 (TaxID: 451515) reference genome sequence is available from NCBI (www.ncbi.nlm.nih.gov).

**Funding:** This research was supported by the Health Research Board (ILP-POR-2019-102 and HRA-POR-2015-1158 to J.P.O'G.), Science Foundation Ireland (19/FFP/6441 to J.P.O'G.), Irish Research Council (GOIPG/2019/2011 to J.P.O'G.), National Institute of Allergy and Infectious Diseases (P01 AI083211 and R01 AI125588 to V.C.T), Swedish Research Council (to F.C.), Laboratory for Molecular Infection Medicine Sweden (to F.C.), Knut and Alice Wallenberg Foundation (to F.C.) and Kempe Foundation (to F.C.). The funders had no role in study design, data collection and interpretation, manuscript writing or the decision to submit the work for publication.

**Competing interests:** The authors have declared that no competing interests exist.

## Author summary

High-level resistance to penicillin-type (β-lactam) antibiotics significantly limits the therapeutic options for patients with MRSA infections necessitating the use of newer agents, for which reduced susceptibility has already been described. Here we report for the first time that the central metabolism pentose phosphate pathway controls MRSA resistance to penicillin-type antibiotics. We comprehensively demonstrated that mutation of the PPP gene *pgl* perturbed metabolism in MRSA leading to increased flux to cell envelope precursors to drive increased antibiotic resistance. Moreover, increased resistance was associated with reduced levels of the lipoteichoic acids in the cell envelope, reduced rates of cell lysis under β-lactam stress and was dependent on the VraRG/GraRS multienzyme membrane complex that controls D-alanylation of teichoic acids and cell surface charge. Our data provide new insights on MRSA mechanisms of β-lactam resistance, which will support efforts to expand the treatment options for infections caused by this and other antimicrobial resistant pathogens.

## Introduction

The World Health Organization (WHO) recently reported a dramatic increase in antimicrobial resistance (AMR) among human pathogens [1, 2]. Exacerbation of the AMR crisis is driven by the misuse and overuse of last-resort antibiotics, the decline in new antimicrobial drugs being approved for clinical use and a lack of mechanistic understanding of AMR in bacterial pathogens [1, 2]. *Staphylococcus aureus*, which is among the most challenging AMR human pathogens, can cause a variety of infections. Skin and soft tissue infections can be localised or enter the vasculature [3, 4], whereas osteomyelitis, septic arthritis, infective endocarditis and pneumonia are deep-seated and systemic [5–13].

Introduction of penicillin to treat *S. aureus* bacteraemia patients in the early 1940s was immediately followed by isolation of penicillin resistant *S. aureus* strains [14]. In *S. aureus*, penicillin resistance is mediated by the β-lactamase enzyme encoded by *blaZ*, which cleaves the β-lactam ring, thus disrupting the activity of the β-lactam antibiotic [14, 15]. Methicillin, a penicillin derivative resistant to β-lactamase hydrolysis, was introduced in 1960s, but was quickly followed by the emergence of methicillin resistant *S. aureus* (MRSA) [16]. Methicillin resistance was driven to the acquisition of the *mecA* gene on *Staphylococcus* cassette chromosome *mec* (SCC*mec*) elements, which encodes an alternative penicillin-binding protein, PBP2a, with a decreased affinity to β-lactams [17–21]. In addition to *mecA*, auxiliary factors also contribute to high-level MRSA β-lactam resistance [22–36], including several involved in the synthesis of cell wall precursors, as well other physiological processes.

The ability of *S. aureus* to adapt to diverse host environments is linked to its ability to obtain essential nutrients from host tissues [37, 38], which in turn is dependent on metabolic reprogramming. A growing body of literature links central metabolic pathways to the pathogenicity of *S. aureus*, from its capacity to proliferate within the host, to the control of antibiotic resistance [22, 37–41]. Thus, the identification of new drug targets and antibacterial strategies is reliant on first understanding virulence mechanisms associated with reprogramming of central metabolic pathways and their role in pathogenesis and antimicrobial resistance.

Bacteria synthesize macromolecules from 13 biosynthetic intermediates derived from glycolysis, the pentose phosphate pathway (PPP) and the tricarboxylic acid (TCA) cycle [42]. *S. aureus* has the complete enzyme set for all three pathways, although it lacks a glyoxylate shunt [42]. In addition to producing pentose precursors for biosynthesis of nucleotides and several

amino acids, the PPP plays a critical role in cellular metabolism, maintaining carbon homeostasis by glucose turnover and contributing to the regeneration of reducing power in the form of NADPH [43–48]. There are two branches in the PPP: the oxidative branch contributes to oxidative stress tolerance by generating reducing power in the form of NADPH/H$^+$, and the non-oxidative branch produces ribose-5-P used in the *de novo* purine synthesis and the generation of nucleotide pools (ATP, ADP, AMP, c-di-AMP, GTP, GDP, GMP, ppGpp, pppGpp, IMP, XMP, etc.) for repair and synthesis of aromatic amino acids and peptidoglycan [47, 48]. PPP activity is increased by environmental stress in Gram-positive organisms [48, 49].

Even though the contribution of glycolysis/gluconeogenesis and the PPP to intracellular persistence of *S. aureus* has been the subject of numerous studies [37, 38, 40, 45, 46, 48, 49], the role of these major glucose metabolism pathways in the antibiotic resistance of *S. aureus* remains largely unstudied. Mutations in PPP enzymes have been previously identified in slow growing-vancomycin intermediate *S. aureus* isolates [50].

We and others have previously reported that purine nucleotide homeostasis plays a key role in the regulation of β-lactam resistance in MRSA [49–53]. Mutations in the *pur* operon and purine salvage pathway were associated with increased resistance, whereas exposure of MRSA to the purine nucleosides guanosine or xanthosine reduced β-lactam resistance [53]. The purine-derived second messenger signalling molecules (p)ppGpp and c-di-AMP regulate β-lactam resistance, and exposure to exogenous guanosine downregulated c-di-AMP levels in *S. aureus* [53].

In this study, we investigated if mutations upstream of purine biosynthesis also control β-lactam resistance focusing on *pgl*, which is the only mutable gene in the oxidative phase of the PPP. We show that a *pgl* mutation in MRSA strain JE2, which leads to a slight growth defect in laboratory growth media, increased β-lactam resistance, but did not cause changes in PBP2a levels or peptidoglycan architecture. Carbon tracing and metabolomics experiments revealed increased flux to glycolysis and several cell envelope precursors. The susceptibility of wild-type JE2 to β-lactam antibiotics was dramatically increased in chemically defined medium containing glucose (CDMG), and accompanied by extensive cell lysis, whereas the *pgl* mutant remained highly resistant, exhibited a thick cell wall, intact septa and had a ruffled cell surface. Lipoteichoic acid (LTA) levels were reduced in the *pgl* mutant and the surface charge of *pgl* cells was significantly increased. β-lactam resistance in the *pgl* mutant reverted to wild-type levels by mutations in the ABC transporter VraFG and cognate two-component regulatory system GraRS. These data reveal that metabolic reprogramming in an MRSA *pgl* mutant increases β-lactam resistance via VraFG/GraRS-dependent changes in cell envelope biogenesis.

## Results

### β-lactam resistance is increased in a MRSA *pgl* mutant

Extrapolating from previous data showing that purine metabolism controls β-lactam resistance [26, 41, 53–56], we turned our attention to the PPP, which produces ribose-5-P, a major substrate for purine and pyrimidine biosynthesis (Fig 1). Given the important role of the PPP in central metabolism and production of reducing power, it is perhaps not surprising that mutations in the key enzymes in this pathway, including *zwf* and *gnd*, are not available in the Nebraska Transposon Mutant library (NTML) [57]. However, the NTML does contain a mutation in the monocistronic *pgl* gene (SAUSA300_1902, NE202), which encodes 6-phosphogluconolactonase, the second enzyme in the oxidative phase of the PPP that converts 6-P-gluconolactone to gluconate-6-P.

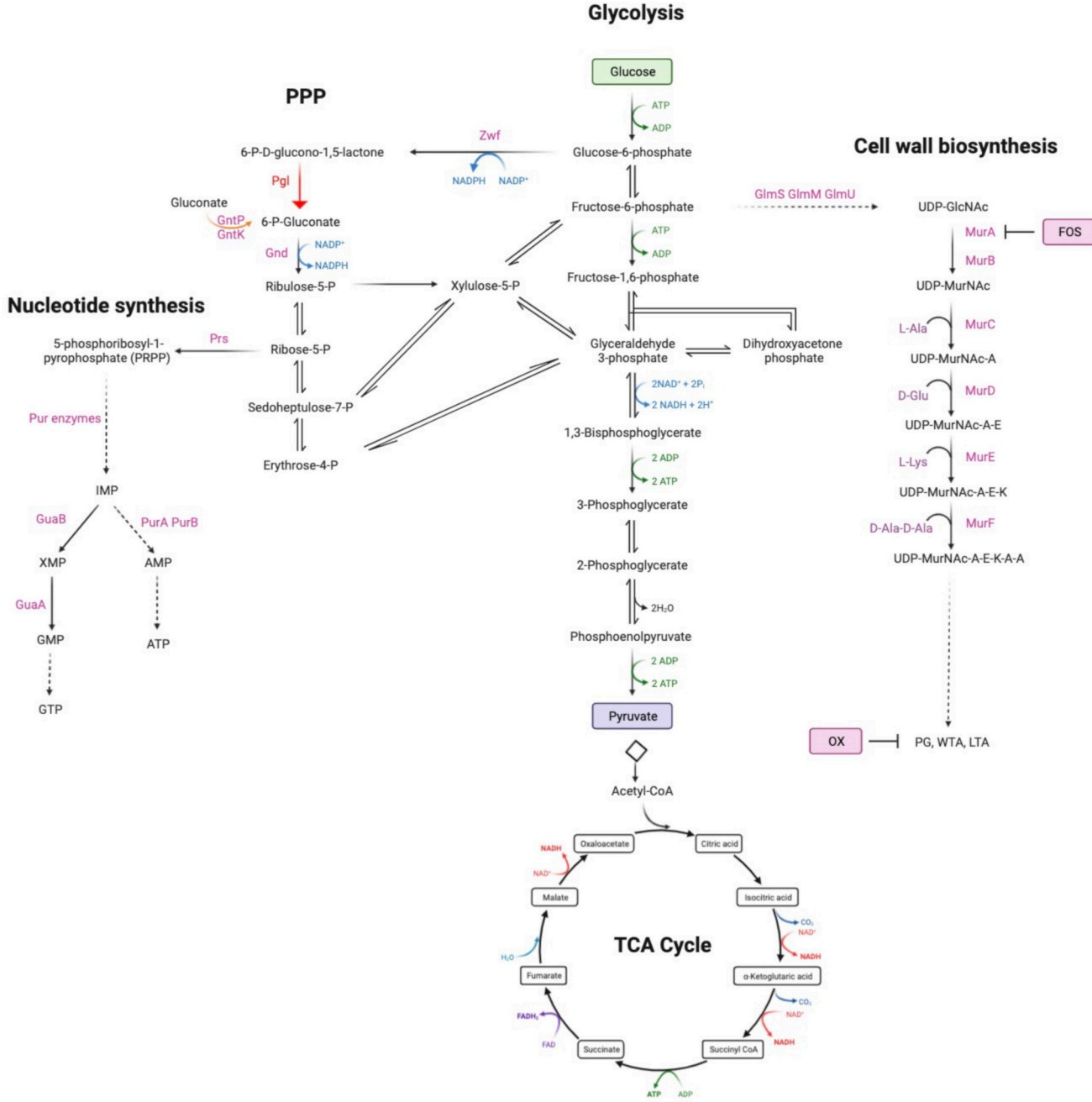

**Fig 1. Summary of the oxidative phase of the pentose phosphate pathway including 6-phosphogluconolactonase (Pgl), that converts 6-P-gluconololactone to gluconate-6-P.** For reference, key glycolysis, TCA cycle, nucleotide and cell wall biosynthetic pathway intermediates are also shown. Fructose-6-P is fluxed from glycolysis to peptidoglycan (PG), wall teichoic acid (WTA) and lipoteichoic acid (LTA) via UDP-N-acetylglucosamine (UDP-GlcNAc) and UDP-N-acetylmuramic acid (UDP-MurNAc). Fosfomycin (FOS) targets MurA which together with MurB is required for the conversion of UDP-GlcNAc to UDP-MurNAc. Oxacillin (OX) targets the transpeptidase activity of the penicillin binding proteins required for PG crosslinking. The putative gluconate shunt involves the export of 6-phosphogluconolactone, which spontaneously degrades to gluconate before being transported into the cell by the gluconate permease GntP and phosphorylated by the gluconate kinase GntK. Schematic made using Biorender.com.

When grown in Mueller Hinton 2% NaCl broth (MHB) the *pgl* mutant NE202 exhibited significantly increased resistance to cefoxitin in disk diffusion assays (zone diameters were 11mm for JE2 versus 8mm for *pgl*) and oxacillin (OX) in broth dilution assays (Table 1).

**Table 1. Minimum inhibitory concentrations (μg/ml; % for Congo Red) of strains used in this study to oxacillin (OX), targocil (TG), tunicamycin (TM), fosfomycin (FOS), D-cycloserine (DCS), Congo Red (CR), vancomycin (VAN), amsacrine (AMS), sulfamethoxazole (SMX) and polymyxin B (PMB) in Mueller Hinton Broth (+ 2% NaCl for OX).**

| Strain | OX | TG | TM | FOS | DCS | CR | VAN | AMS | SMX | PMB |
|---|---|---|---|---|---|---|---|---|---|---|
| JE2 | 64 | 1–2 | 4 | 32–64 | 16–32 | 0.25% | 1 | >256 | 128–256 | 128 |
| *pgl* | 128–256 | 4–8 | 4 | 64–128 | 32 | 0.03% | 2–4 | 32–64 | 16–32 | 64 |
| *pgl*comp | 64 | 1–2 | 4 | 32–64 | 32 | 0.25% | 1–2 | >256 | 128–256 | n/d |
| *pgl*::Km$^r$ | 128–256 | 4–8 | 2–4 | 64–128 | 32 | n/d | 2–4 | n/d | 16–32 | n/d |
| *pgl*/*mecA* | 0.5 | n/d | n/d | n/d | n/d | n/d | n/d | n/d | n/d | n/d |
| *mecA* | 0.25 | n/d | n/d | n/d | n/d | n/d | n/d | n/d | n/d | n/d |
| JE2 *pgl*::tn | 128–256 | n/d | n/d | n/d | n/d | n/d | n/d | n/d | n/d | n/d |
| *vraG* | 64 | n/d | n/d | n/d | n/d | n/d | 0.5 | n/d | n/d | 2–4 |
| *vraF* | 64 | n/d | n/d | n/d | n/d | n/d | n/d | n/d | n/d | n/d |
| *pgl*/*vraG* | 128–256 | n/d | n/d | n/d | n/d | n/d | 0.5 | n/d | n/d | 4 |
| *pgl*/*vraF* | 128–256 | n/d | n/d | n/d | n/d | n/d | n/d | n/d | n/d | n/d |

n/d—not determined

Comparative whole genome sequencing analysis confirmed the absence of unexpected secondary mutations outside the *pgl* locus in NE202. The NE202 phenotype was verified by (i) showing that increased OX resistance was acquired by wild-type following transduction of the *pgl*::Erm$^r$ allele and (ii) complementation of NE202 with the wild-type *pgl* gene (*pgl*comp) (Table 1). A *pgl*/*mecA* mutant was OX susceptible (Table 1) and Western immunoblotting revealed no differences in PBP2a expression between wild-type JE2, *pgl* and *pgl*comp grown in TSB supplemented with OX 0.5 μg/ml (S1A Fig) or MHB 2% NaCl supplemented with OX 32 μg/ml (S1B Fig). Thus, high-level OX resistance in *pgl* was dependent on *mecA* but was not associated with increased PBP2a expression.

## The *pgl* OX resistance phenotype is glucose-dependent and unrelated to changes in peptidoglycan (PG) structure

Colonies of *pgl* were smaller than JE2 on MHA plates (S2A Fig) and, in the absence of antibiotics, the *pgl* mutation negatively impacted growth in MHB (S2B Fig), but to a lesser extent in LB, TSB and BHI (S2C–S2E Fig). A *pgl* growth defect was also measured in chemically defined media with glucose (CDMG), but not in CDM without glucose (S2F and S2G Fig). Growth of the complemented *pgl*comp mutant was indistinguishable from the wild-type JE2 under all culture conditions tested (S2B–S2F Fig). The mild growth defects of *pgl* in MHB and CDMG correlated with significantly increased OX MICs (Table 1, Fig 2A), whereas the MIC of *pgl* in CDM (32–64 μg/ml) was more similar to wild-type JE2 (16–32 μg/ml; Fig 2A). Notably, not only was *pgl* more resistant than wild-type JE2 in CMDG, but wild-type JE2 OX resistance was significantly reduced in this growth medium (MIC = 0.5–1 μg/ml; Fig 2A). Wild-type JE2 and *pgl* grew similarly in CDM OX 10 μg/ml (Fig 2B), whereas only *pgl* was able to grow in CDMG OX 10 μg/ml (Fig 2C). Doubling dilutions of the CDMG glucose concentration from 5 g/l (28 mM) further revealed that even at the lowest glucose concentration tested, 0.07 g/l (0.43 mM), JE2 growth in OX was significantly impaired (S3A Fig). Similarly, the OX MIC of JE2 was reduced from 32–64 to ≤1 μg/ml in CDMG at glucose concentrations >0.3 g/l (1.75 mM) (S3B Fig). These data indicate that mutation of *pgl* significantly increased the glucose-dependent reduction of OX resistance in JE2 at physiologically relevant concentrations. Unlike wild-type JE2, the *pgl* mutant was able to grow at OX concentrations >1 μg/ml in 25 or 70% human serum (S4 Fig), and exhibited a higher OX MIC in CDMG or MHB supplemented with up to

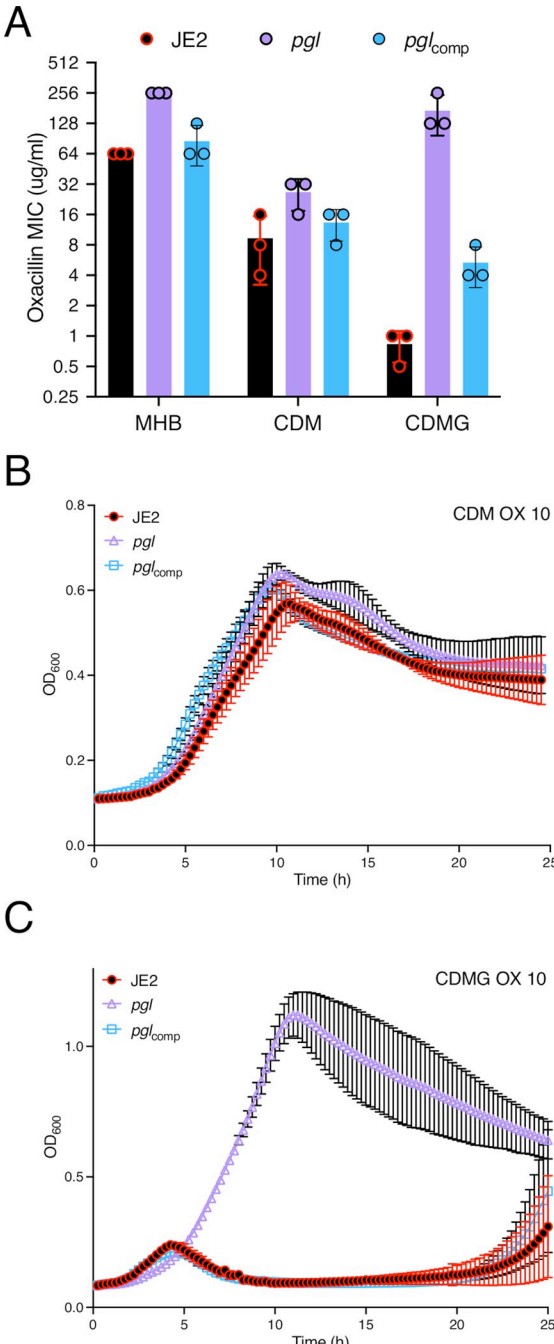

**Fig 2. Mutation of *pgl* increases resistance to oxacillin. A.** Oxacillin MICs of JE2, *pgl* and the complemented *pgl* mutant in Mueller Hinton broth with 2% NaCl (MHB), chemically defined media (CDM) and CDM with glucose (CDMG). Note that the Y axis (Oxacillin MIC) is a log2 scale. **B and C.** Growth of JE2, *pgl* and *pgl*comp for 25 hrs at 35°C in CDM (B) and CDMG (C) supplemented with OX 10 μg/ml. Growth ($OD_{600}$) was measured at 15 min intervals in a Tecan plate reader. Data are the average of 3 independent experiments and error bars represent standard deviation.

70% human serum (S1 Table) further demonstrating the *in vivo* relevance of this phenotype. The *pgl* mutation increased sensitivity to oxidative stress ($H_2O_2$) in CDMG (S5 Fig), similar to previous observations in *Listeria monocytogenes* using BHI media [58].

Confocal microscopy revealed that the diameter of *pgl* cells from overnight CDMG grown cultures was significantly smaller than wild-type JE2 or *pgl*$_{comp}$ cells (Fig 3A,B). The significant OX-induced increase in MRSA cell size, which we and others have previously reported [31, 53, 59–61], was more pronounced in wild-type JE2 and *pgl*$_{comp}$ than the *pgl* mutant (Fig 3C). Furthermore, the increased cell size of wild-type JE2 and *pgl*$_{comp}$ in CDMG OX was associated with a dramatic increase in the number of cells undergoing visible lysis (Fig 3D), an observation consistent with the abrupt decline in the OD$_{600}$ of wild-type JE2 and *pgl*$_{comp}$ cultures after 4–5 h growth under these growth conditions (Fig 2C). Quantitative PG compositional analysis of muramidase-digested muropeptide fragments revealed similar oligomerisation profiles and crosslinking for wild-type JE2, *pgl* and the *pgl*$_{comp}$ strains grown in CDMG, or CDMG supplemented with sub-inhibitory 0.05 μg/ml OX, MHB 2% NaCl, MHB 2% NaCl supplemented with 0.5 μg/ml OX (S6A–S6D Fig). The total PG content was also similar for all three strains under these growth conditions (S6E–S6H Fig). Thus, in addition to the unchanged PBP2a expression (S1 Fig), increased *pgl* OX resistance was unrelated to changes in PG structure or amount (S6 Fig).

## Exogenous D-gluconate or mutation of the *gntPK* gluconate shunt genes did not restore wild-type OX resistance in the *pgl* mutant

In *Escherichia coli* and *L. monocytogenes*, 6-phosphogluconolactone that accumulates in *pgl* mutants is dephosphorylated to labile gluconolactone, which is exported out of the cell where it spontaneously hydrolyses to gluconate [58, 62]. In *S. aureus*, the predicted gluconate shunt genes *gntP* (SAUSA300_2442) and *gntK* (SAUSA300_2443) are co-located on the chromosome with the *gntR* regulator. In a previous RNAseq analysis, we reported that *gntP* was upregulated by OX [63]. Growth and OX resistance of wild-type JE2, *pgl* and *pgl*$_{comp}$ were similar in CDMG supplemented with 5 g/l D-gluconate and CDMG OX D-gluconate (S7A and S7B Fig). Inactivation of *gntP* or *gntK* in the *pgl* mutant was accompanied by a modest growth delay in CDMG OX but did not restore wild-type levels of OX susceptibility (S7C Fig). Therefore, exogenous D-gluconate and the gluconate shunt genes are not notably involved in the increased OX resistance phenotype of the *pgl* mutant.

## Inactivation of *pgl* reduces carbon flux through PPP

Liquid chromatography-tandem mass spectrometry analysis was used to trace [1,2-$^{13}$C$_2$] glucose flux through glycolysis and the PPP in wild-type JE2, *pgl* and *pgl*$_{comp}$. As described previously [64], six-carbon [1,2-$^{13}$C] glucose can be metabolised via glycolysis and the PPP to produce three-carbon $^{13}$C$_2$-pyruvate (M+2) and $^{13}$C$_1$-pyruvate (M+1), respectively (Fig 4A). The M+1 fraction is produced following a decarboxylation reaction in the PPP that releases $^{13}$CO$_2$ (Fig 4A). M+1 pyruvate levels were reduced in *pgl*, indicative of reduced PPP activity (Fig 4B), whereas M+2 pyruvate levels derived primarily from glycolysis, were similar (Fig 4B). The M+2/M+1 ratio further illustrated the impaired PPP activity of *pgl* and showed >2-times more pyruvate generated directly from glucose entering glycolysis in *pgl* than in wild-type JE2 or *pgl*$_{comp}$ (Fig 4C).

## OX resistance in the *pgl* mutant is independent of the TCA cycle or glucogenic and ketogenic amino acids

HPLC was used to investigate if redirected glucose flux from the PPP to glycolysis impacted consumption of amino acids in CDMG, and revealed that levels of threonine, the branched chain amino acids (BCAAs) valine, leucine and isoleucine, as well as phenylalanine,

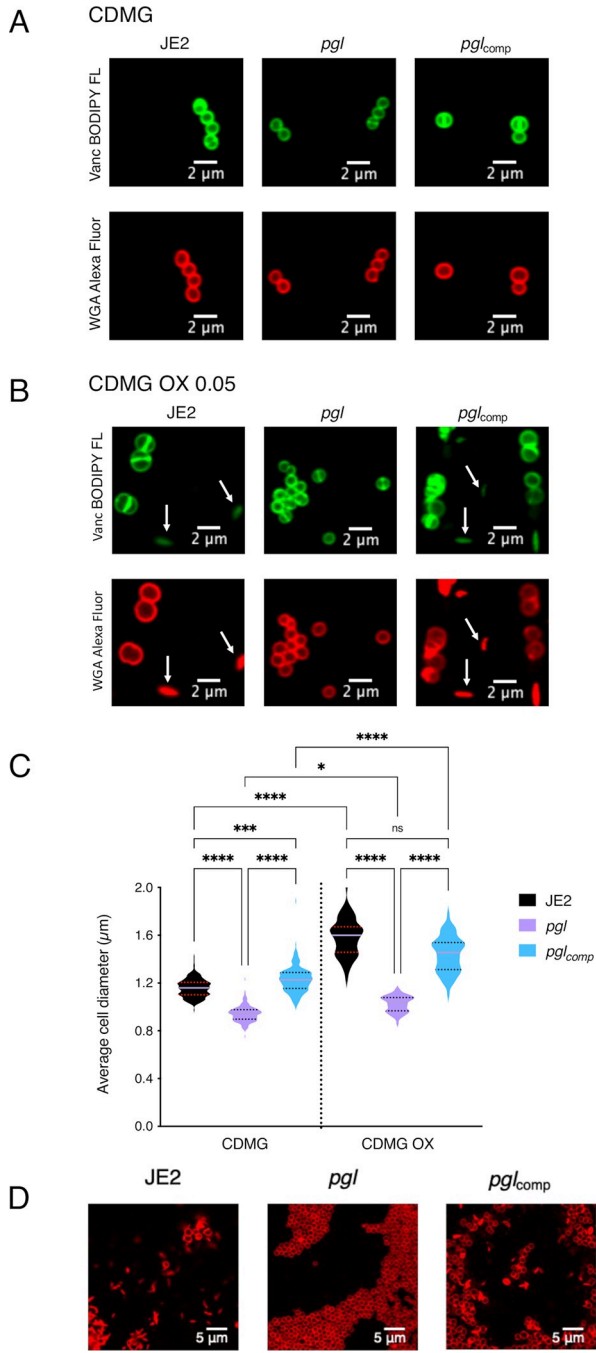

**Fig 3. Mutation of *pgl* reduces cell size and prevents OX-induced cell lysis in CDMG. A and B.** Representative microscopic images of JE2, *pgl* and *pgl*comp cells grown in CDMG (A) or CDMG supplemented with OX 0.05 μg/ml (B) and labelled with vancomycin BODIPY FL, which binds to the terminal D-ala-D-ala in the peptidoglycan stem peptide (green, top panel) or WGA Alexa Fluor 594, which binds to GlcNAc and other sugars in the cell envelope (red, bottom panel). **C.** Average diameter of JE2, *pgl* and *pgl*comp cells grown in CDMG or CDMG OX. Images of cells from four biological replicates were acquired using Fv3000 confocal microscope and software, 50 cells measured per biological replicate (200 cells in total) for CDMG and 60 cells in total counted for CDMG OX (due to cell lysis), and the violin plots for the four biological replicates were generated using GraphPad Prism V9. Asterisks indicate statistically significant difference according to using a Kruskal-Wallis test followed by a Dunn's multiple comparison test. Adjusted p-values * $p < 0.05$, *** $p < 0.001$ and **** $p < 0.0001$ are indicated. **D.** Extensive lysis of JE2 and *pgl*comp (but not *pgl*) in CDMG OX 0.05 μg/ml cultures. Cells were labelled with WGA Alexa Fluor 594 and representative microscopic images are shown.

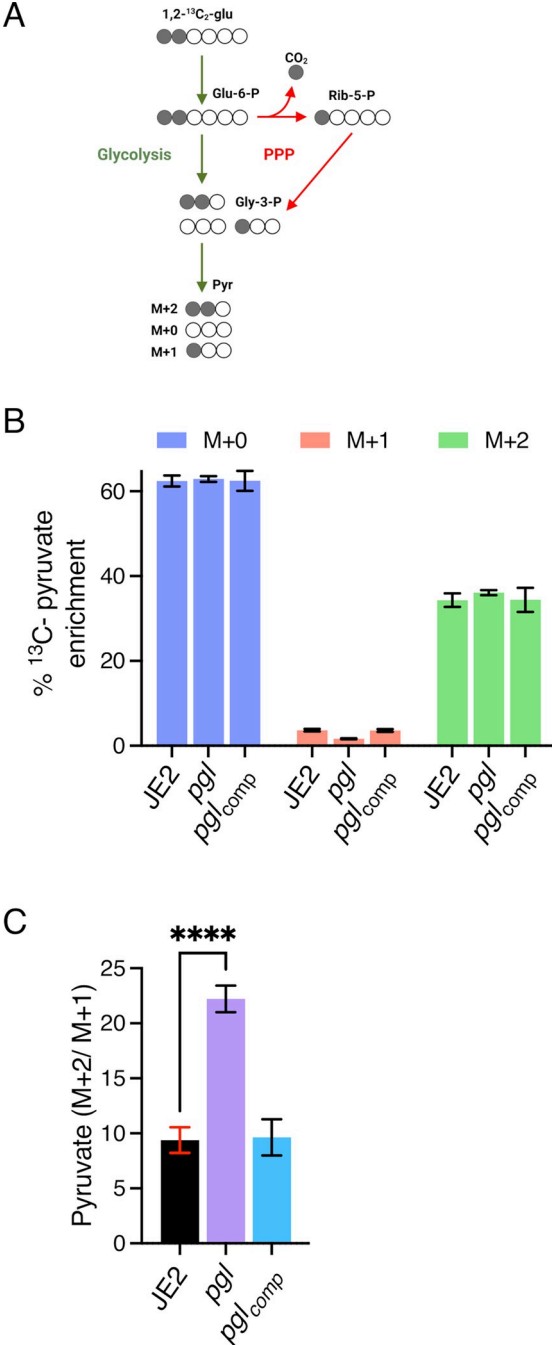

**Fig 4. PPP activity is impaired in the *pgl* mutant. A.** JE2, *pgl* and the complemented *pgl* mutant (*pgl*comp) were grown in CDM [1,2-$^{13}$C]Glucose and fluxes via glycolysis and the pentose phosphate pathway (PPP) were compared as described previously [64]. The M+2 pyruvate is unique to glycolysis and the M+1 pyruvate to PPP. Thus, the M+2/M+1 ratio is indicative of carbon flux through glycolysis relative to PPP. The M+0 pyruvate can arise from different sources including the unlabeled part of the [1,2-$^{13}$C]Glucose and pyruvogenic amino acids that are consumed alongside glucose. **B.** Relative levels of M+1 pyruvate indicative of PPP activity and M+2 pyruvate indicative of glycolytic activity in JE2, *pgl* and *pgl*comp. **C.** The M+2/M+1 ratio indicative of pyruvate produced directly from glucose flux through glycolysis in JE2, *pgl* and *pgl*comp. Data are the average of three independent experiments and standard deviations are shown. Significant differences were determined using ordinary one-way ANOVA with Dunnett's multiple comparison using GraphPad Prism V9 and adjusted p-value **** $p < 0.0001$ is indicated.

tryptophan and tyrosine, histidine, methionine and aspartic acid were increased in the super-natant of *pgl* cultures compared to JE2 or *pgl*comp after 7.5 h growth (S8A Fig). Interestingly, the levels of the TCA cycle intermediates malate, succinate and particularly α-ketoglutarate were also increased in CDMG supernatants of *pgl* (S8B Fig), which may be consistent with a reduced requirement for glucogenic and ketogenic amino acids. To investigate this proline dehydrogenase (*putA*::Erm^r) and glutamate dehydrogenase (*gudB*::Erm^r) mutations, predicted to interfere with the flux of amino acids to α-ketoglutarate, were transduced from the NTML [57] into *pgl*::Km^r. Growth of the resulting *pgl*/*putA* and *pgl*/*gudB* mutants in CDMG and CDMG OX was similar to *pgl*::Km^r (S8C and S8D Fig). Similarly the *pgl* TCA cycle double mutants *pgl*/*sdhA*, *pgl*/*sucA* and *pgl*/*sucC* remained capable of growing in CDMG OX (S8C and S8D Fig), although *pgl*/*sucC* exhibited an extended lag phase in keeping with our previous report that *sucC* mutation re-sensitizes MRSA to β-lactam antibiotics due to increased accu-mulation of succinyl CoA [39]. Collectively, these data indicate that an intact TCA cycle or the accumulation of TCA cycle intermediates and ketogenic amino acids in culture supernatants was not associated with the increased β-lactam resistance of the *pgl* mutant.

## Increased resistance to β-lactam antibiotics in *pgl* is promoted by redirected carbon flux to cell wall precursors

Whole cell metabolomics was performed on JE2, *pgl* and *pgl*comp grown in CDMG or CDMG OX (Fig 5). Consistent with the important role of the PPP in the generation of reducing power and nucleotide biosynthesis, levels of key redox carriers and six nucleotides were significantly reduced in *pgl* and restored to JE2 levels in the complemented mutant (Fig 5). Interestingly, reduced nucleotide levels correlated with a 2-4-fold increase in the susceptibility of *pgl* mutant to sulfamethoxazole, which inhibits dihydropteroate synthetase in the folate synthesis pathway (Table 1). Levels of sedoheptulose 7-P which is downstream of Pgl in the PPP was also reduced in *pgl*, reaching significance in CDMG, whereas ribose 5-P and erythrose 5-P were significantly increased (Fig 5), indicative of complex metabolic reprogramming in the *pgl* mutant.

Consistent with the [1,2-$^{13}$C] glucose tracing experiments, accumulation of fructose 6-P from which cell wall precursors are derived, was increased in CDMG OX and significantly increased in CDMG (Fig 5). Furthermore, the downstream glycolytic intermediates fructose 1,6-bis-P, dihydroxyacetone phosphate (DHAP), glyceraldehyde 3-P and phosphoenolpyr-uvate (PEP) were reduced (Fig 5). Although there are several possible explanations for this, one possibility is that the accumulated fructose 6-P may be fluxed to the PPP or cell wall. Indeed, significantly increased levels of UDP-mono and UDP-penta were measured in *pgl* grown in CDMG OX, but not in CDMG (Fig 5). In contrast, the levels of UDP-GlcNAc and UDP-MurNAc were significantly decreased (Fig 5), perhaps reflecting increased consumption of these substrates in the production of UDP-mono and UDP-penta in CDMG OX. Increased accumulation of UDP-mono and UPD-penta correlated with the increased resistance of the *pgl* mutant to fosfomycin (FOS) (Table 1, S9 Fig), an antibiotic that inhibits the MurA enzyme, which together with MurB catalyses the conversion of UDP-GlcNAc to UDP-MurNAc (Fig 1). Furthermore, *pgl* exhibited significantly increased resistance to an OX/FOS combination com-pared to wild-type JE2 in a checkerboard dilution assay (S9 Fig). Broth microdilution suscepti-bility experiments revealed that the *pgl* mutant was 1-2-fold more resistant to vancomycin (VAN), which targets the terminal D-ala-D-ala of the PG stem peptide (Table 1).

Taken together, these data indicate that redirected carbon flux to cell wall precursors in *pgl* contributes to the increased resistance to β-lactam antibiotics. Furthermore, *pgl* viability appears to be underpinned by a complex and regulated interconversion of glycolytic and PPP

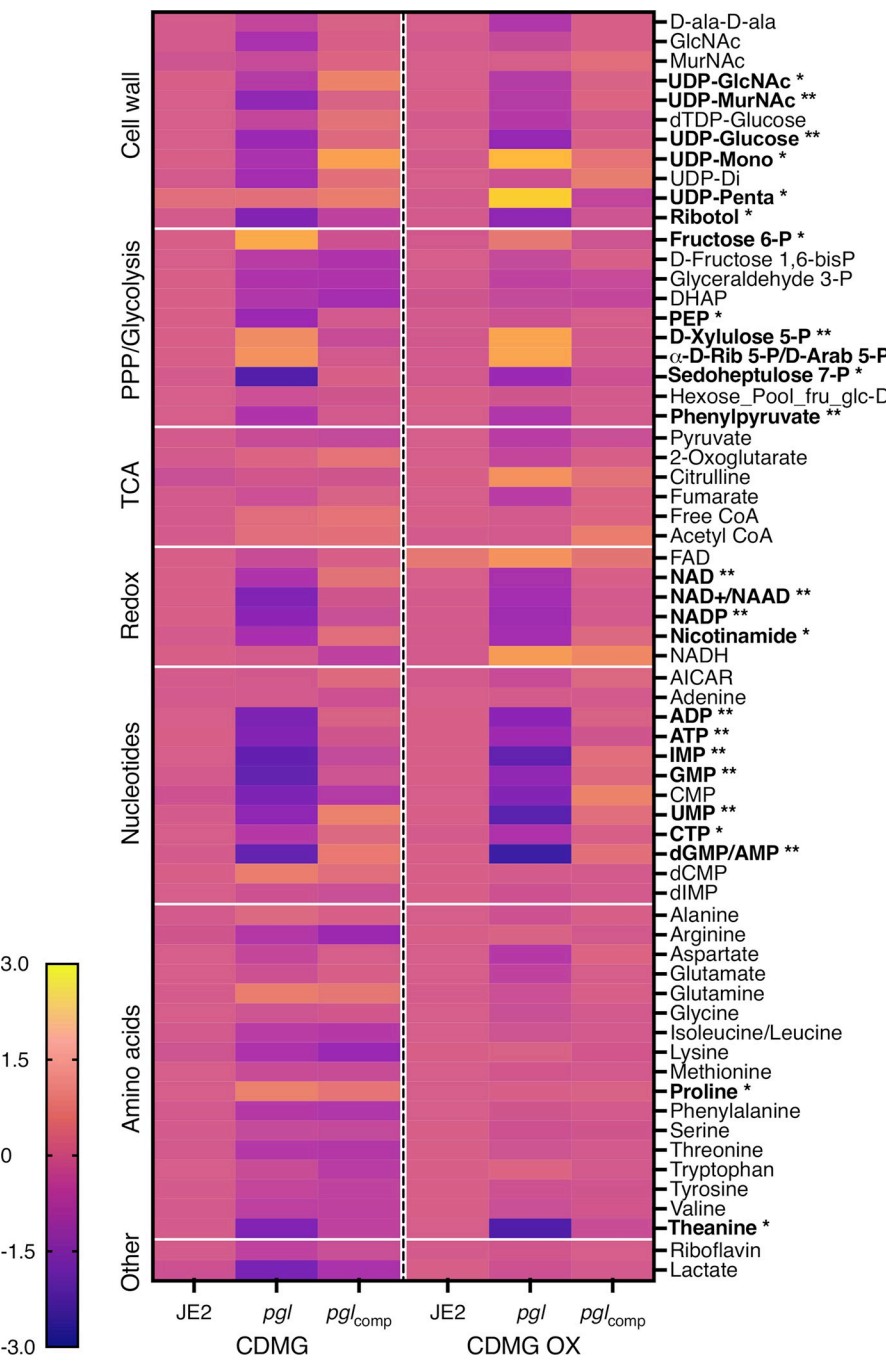

**Fig 5. Heatmap comparison of cell wall, pentose phosphate pathway (PPP)/glycolysis, TCA cycle, redox, nucleotides and amino acid metabolites in JE2, *pgl* and *pgl*comp.** Whole cell metabolomics was performed on JE2, *pgl* and *pgl*comp grown in CDMG and CDMG OX 10 μg/ml and the cells collected after 4–5 hours (early exponential phase). Data presented are the average of three biological replicates (2 biological replicates for FAD) analysed using GraphPad Prism V9. Individual metabolite levels that were significantly different using a one-way ANOVA with Turkey's post-hoc in *pgl* grown in CDMG, CDMG OX or both are highlighted in bold text. * significant difference in either CDMG or CDMG OX. ** significant difference in both CDMG and CDMG OX.

intermediates, which may also explain why the glycolytic shunt genes are dispensable for the growth of the *pgl* mutant under these culture conditions.

## Mutation of *pgl* alters susceptibility to antimicrobial agents targeting wall teichoic acids (WTAs) and lipoteichoic acids (LTAs) and is accompanied by morphological changes in the cell envelope

The MICs of wild-type JE2 and *pgl* to the TarO inhibitor tunicamycin were the same, whereas *pgl* was more resistant to the TarGH inhibitor targocil and more susceptible to the D-alanylation inhibitor amsacrine (Table 1), revealing different effects of antimicrobial agents targeting distinct steps in WTA biosynthesis. TarO catalyzes the transfer of N-acetylglucosamine-1-phosphate from UDP-GlcNAc to undecaprenyl-P to initiate WTA synthesis [65]. The TarGH ABC transporter transports WTAs across the cytoplasmic membrane [66], and the polymer is then D-alanylated by the DltABCD complex [67]. The *pgl* mutant was also more sensitive to Congo Red which targets the LTA synthase LtaS [68] (Table 1). Importantly LTA is also D-alanylated by DltABCD. The susceptibility of *pgl* to D-cycloserine, which targets the alanine racemase and ligase enzymes in the D-ala-D-ala pathway was unchanged when compared to wild-type, and the metabolomic analysis also showed no significant differences in the levels of D-ala-D-ala in wild-type JE2, *pgl* and *pgl*comp (Fig 5).

Transmission electron microscopy (TEM) revealed that *pgl* cells grown in CDMG OX had visibly ruffled surface characteristics, and thick, intact septa compared to JE2 cells (Fig 6). Consistent with previous microscopic analysis (Fig 3), TEM revealed defective/truncated septa in dividing wild-type cells, as well as cells undergoing lysis (Fig 6). In contrast wild-type and *pgl* cells grown in the absence of OX were largely similar (S10 Fig). Taken together these data suggest that cell envelope changes in the *pgl* mutant are the result of altered activity of the TarGH, LtaAS-YpfP and DltABCD membrane complexes involved in export and D-alanylation of WTAs and LTAs that collectively contribute to increased OX resistance.

## OX resistance in the *pgl* mutant is dependent on *vraF* and *vraG*

During experiments to remove the Erm[r] marker in NE202 a *pgl* markerless transposon mutant that had reverted to wild-type patterns of growth, particularly in CDMG OX, was isolated (Fig 7A and 7B). Whole genome sequencing of this mutant, designated *pgl*R1, identified a thymine nucleotide deletion 73bp upstream of *putA* and a Gln394STOP substitution in VraG. Construction of *pgl* double and triple mutants revealed that the *pgl* OX resistance phenotype was dependent on *vraG* and not *putA* (Fig 7A and 7B). *vraG* encodes a membrane permease, which together with its cognate ATPase, VraF, comprises an ABC efflux pump previously implicated in resistance to cationic antimicrobial peptides, polymyxin B and vancomycin [69–73], potentially via the export of cell wall/teichoic acid precursors or modifying subunits [71]. Consistent with this, a *pgl*/*vraF* mutant grew similarly to *pgl*/*vraG* and JE2 in CDMG and CDMG OX (Fig 7A and 7B). VraFG is also part of a multicomponent complex with the glycopeptide resistance-associated GraRS two component system, that regulates *vraFG* as well as the *dltABCD* operon and *mprF* [69–71]. A *pgl*/*graR* mutant exhibited an intermediate phenotype when grown in CDMG OX compared to JE2 and the *pgl*/*vraF* or *pgl*/*vraG* mutants (Fig 7B) revealing a role for the entire VraFG/GraRS complex in the *pgl* OX resistance phenotype. In contrast, consistent with published data [74], mutation of *vraG* was associated with significantly increased susceptibility to polymyxin B, independent of *pgl* (Table 1).

Interestingly, the increased VAN MIC of the *pgl* mutant grown in MHB was reduced from 2–4 μg/ml to 0.5 μg/ml in *pgl*/*vraG* (Table 1) further indicating a reversal of cell envelope

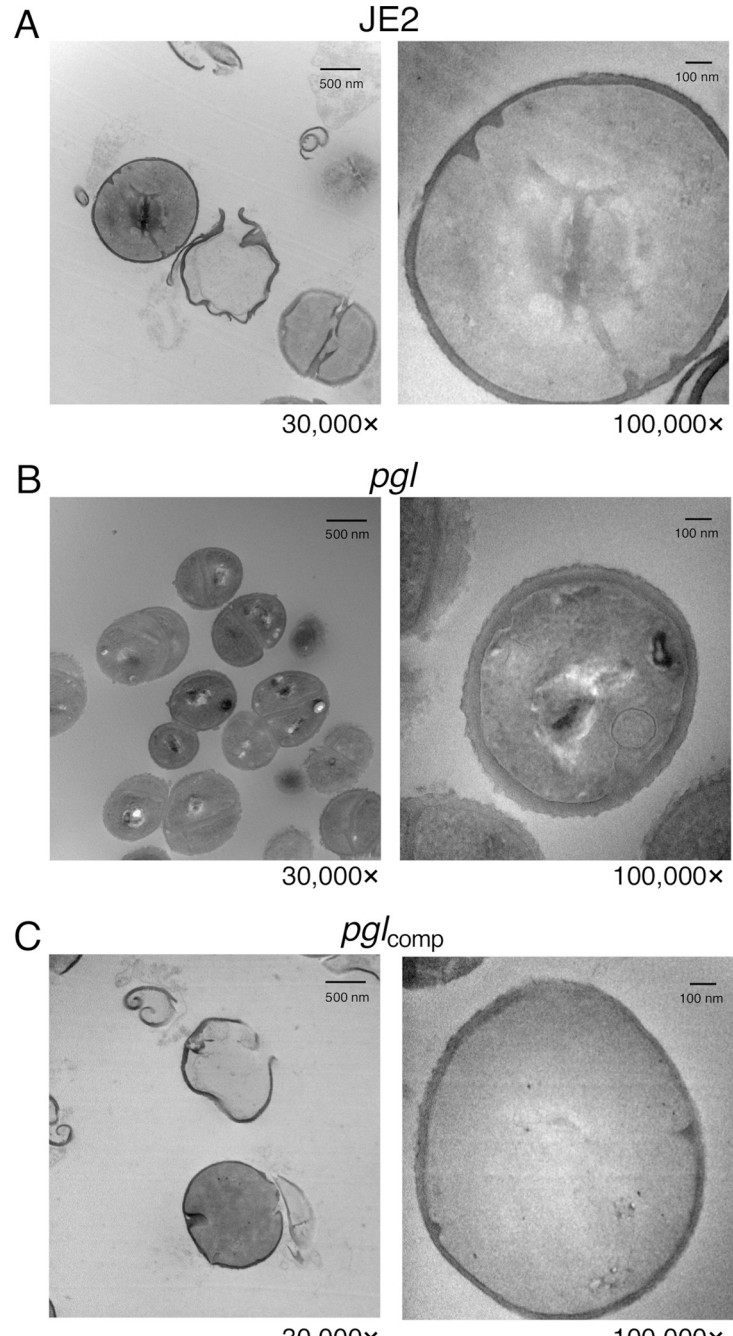

**Fig 6. Increased OX resistance in the *pgl* mutant is associated with a ruffled surface morphology, a thicker cell wall and thicker septa between dividing cells.** Transmission electron microscopy at 30,000× (left) and 100,000× (right) magnification was performed on JE2 (A), *pgl* (B) and *pgl*comp (C) cells collected from exponential phase cultures grown for 4.5 h in CDMG OX 1 μg/ml normalized to $OD_{600}$ = 1 in PBS before being fixed and thin sections prepared. Representative cells from each strain are shown. Scale bars represent 500 nm at 30,000× or 100 nm at 100,000× magnification.

changes. However, *pgl/vraG* and *pgl* cells were the same size in CDMG and CDMG OX (Fig 7C and 7D) demonstrating that the *vraG* mutation does not reverse other central metabolism-related phenotypes in the *pgl* mutant.

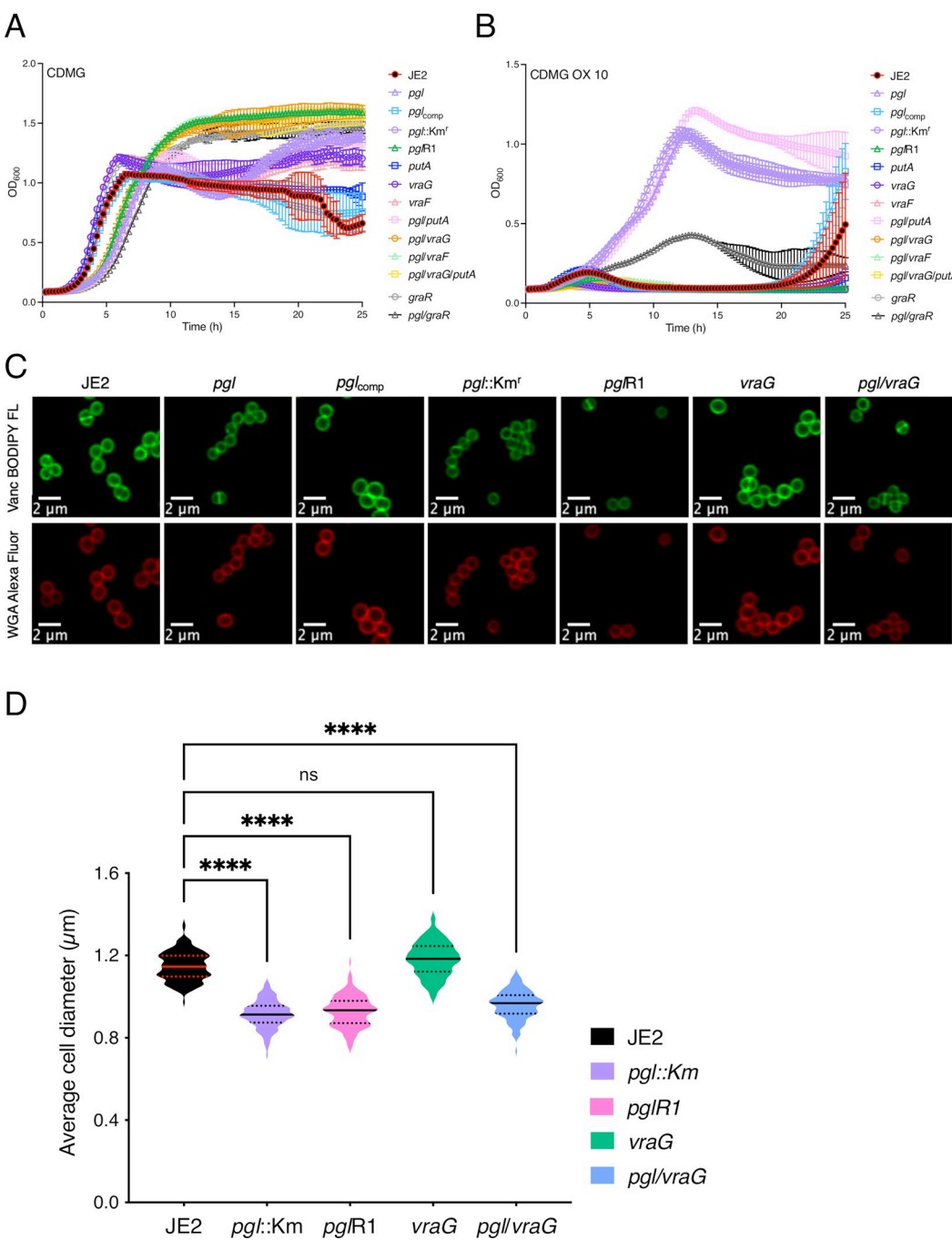

**Fig 7. Mutation of *vraG* restores wild-type OX resistance, but not cell size, in the *pgl* mutant grown in CDMG. A and B.** Growth of JE2, *pgl*::Km[r], *pgl*R1, *putA*, *graR*, *vraG*, *pgl/putA*, *pgl/graR*, *pgl/vraG*, *pgl/putA/vraG* for 25 hrs at 35°C in CDMG (A) and CDMG supplemented with OX 10 μg/ml (B). Growth ($OD_{600}$) was measured at 15 min intervals in a Tecan plate reader. Data are the average of 3 independent experiments and error bars represent standard deviation. **C.** Representative microscopic images of JE2, *pgl*, *pgl*comp, *pgl*::Km[r], *pgl*R1, *vraG* and *pgl/vraG* cells grown in CDMG and labelled with vancomycin BODIPY FL (green, top panel) or WGA Alexa Fluor 594 (red, bottom panel). **D.** Average diameter of JE2, *pgl*::Km[r], *pgl*R1, *vraG* and *pgl/vraG* cells grown in CDMG. Images of cells from three biological replicates were acquired using Fv3000 confocal microscope and software, 50 cells measured per biological replicate (150 cells in total) and the violin plots for the three biological replicates were generated using GraphPad Prism V9. Asterisks indicate statistically significant difference according to using a Kruskal-Wallis test followed by a Dunn's multiple comparison test. Adjusted p-values **** p<0.0001 or ns, not significant are indicated.

In CDMG, the OX MICs of *putA*, *vraF*, *vraG* and *pgl/putA* were the same as JE2, whereas *pgl/graR* was reduced to 32–64 µg/ml and *pgl/putA/vraG*, *pgl/vraF* and *pgl/vraG* were reduced to 8 µg/ml compared to 128–256 µg/ml for *pgl* and *pgl*::Km$^r$ (S11 Fig). Interestingly, in MHB 2% NaCl, the increased OX MIC of the *pgl* mutant (128–256 µg/ml) was not reversed in the *pgl/vraG* or *pgl/vraF* mutants (Table 1), underlining the importance of exogenous glucose in *pgl*-related phenotypes and indicating that VraFG-dependent OX resistance in the *pgl* mutant is environmentally regulated.

## Lipoteichoic acids are reduced in the *pgl* mutant

Levels of teichoic acids in the *pgl*, *vraF*, *vraG* and *graR* strains were compared using several assays. Binding of wheat germ agglutinin (WGA) Alexa Fluor 594 to GlcNAc and other sugars in WTA, LTA and PG was significantly reduced in *pgl* and *pgl*::Km$^r$ compared to the JE2, *vraF*, *vraG* and *graR* strains (Fig 8A). Furthermore, WGA binding was restored to wild-type levels in *pgl*R1, *pgl/vraF*, *pgl/vraG*, *pgl/putA/vraG* and *pgl/graR* (Fig 8A). Levels of ribitol, the backbone for WTAs and LTAs, were also reduced in *pgl* grown in CDMG OX and significantly reduced in CDMG (Fig 5). Levels of WTAs visualised on Alcian blue stained WTA gels were similar in all strains grown in CDMG (Fig 8B). In CDMG OX, WTAs in the *pgl* mutant were marginally reduced in total amount compared to JE2 (Fig 8B), but ImageJ densitometry analysis from 3 independent experiments revealed that this was not significant. Analysis of LTA immunoblots by ImageJ densitometry revealed a significant reduction (p <0.05) in the relative levels of LTAs in *pgl* and *pgl/vraG* compared to JE2 (Fig 8C). The reduced levels of LTAs in *pgl* correlated with the reduction in WGA binding (Fig 8A) and the increased susceptibility of the *pgl* mutant to Congo Red (Fig 8D), which targets LtaS. LTA accumulates at the site of cell division [75], and reduced levels of this glycopolymer may contribute to the defective cell division and reduced cell size phenotypes of *pgl* (Fig 6). Furthermore, because high levels of LTA have been implicated in increased cell lysis [76], these data may also explain in part the increased susceptibility of the wild-type to lysis in CDMG OX compared to *pgl*.

## Evidence that VraG-dependent control of cell surface charge is altered in *pgl*

The reduced LTA levels in the *pgl* mutant raise questions about the impact of the altered WTA:LTA ratio on the *pgl* cell envelope and/or the post-translational modification(s) of teichoic acids (TAs). WTA glycosylation, which has previously been implicated in increased OX resistance [63, 77], was ruled out because the OX MIC of the *pgl* mutant was unaffected by mutations in the WTA α and β glycosylase genes *tarM* and *tarS* (S11 Fig). Because TAs contribute to cell surface charge, cyctochrome c binding assays [74] were performed and revealed that *pgl* cells were significantly more positively charged in CDMG OX (Fig 8E). A similar trend was observed in CDMG, but did not reach significance. Consistent with published data [74], the *vraG* mutant was more negatively charged than wild-type, albeit not reaching significance (Fig 8E). The *vraG* mutation partially restored the surface charge of the *pgl* mutant. Together with the reduction in LTA levels, the increased positive charge is further evidence of the altered composition and decoration of the cell envelope in *pgl*. Moreover, these observations are consistent with increased VraFG/GraRS-dependent, DltABCD-mediated D-alanylation of TAs and increased OX resistance in *pgl*. Future studies to characterise VraFG/GraRS-controlled WTA/LTA polymer length and decoration [76, 78, 79], and patterns of LTA release [27, 80], in *pgl* will be informative. In summary, the data presented here reveal that extensive metabolic reprogramming in a MRSA *pgl* mutant is accompanied by increased OX resistance, which is associated with redirected carbon flux to cell envelope precursors, reduced levels of

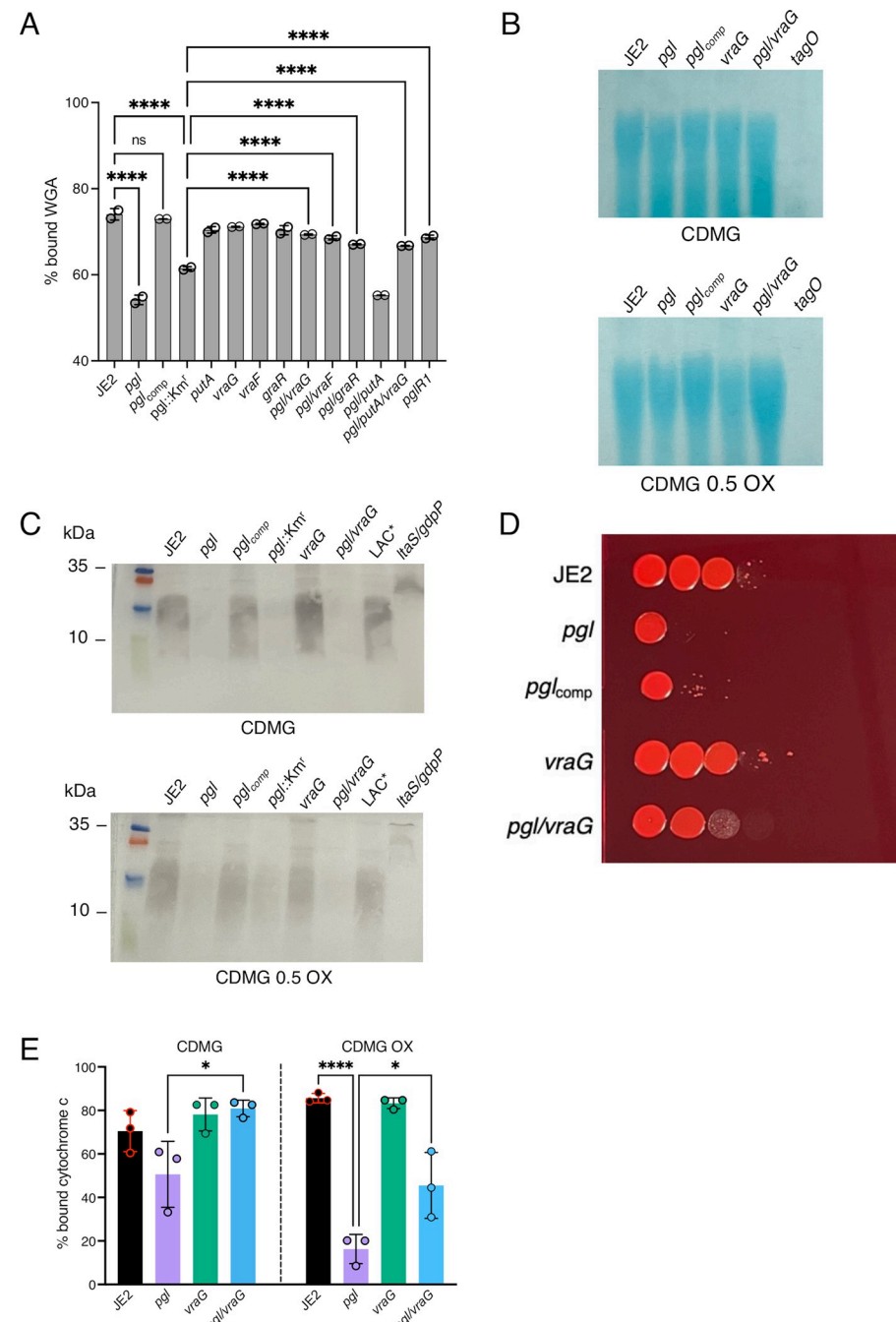

**Fig 8. Lipoteichoic acids are reduced and the cell surface is more positively charged in the *pgl* mutant. A.**
Comparison of wheat germ agglutinin (WGA) Alexa Fluor 594 binding to JE2, *pgl*, *pgl*comp, *pgl*::Km$^r$, *vraG*, *vraF*, *graR*,
*putA*, *pgl/vraG*, *pgl/vraF*, *pgl/graR*, *pgl/putA*, *pgl/putA/vraG and pgl*R1 cells grown for 4.5 h in CDMG OX using
fluorescence microscopy at 594nm excitation/618nm detection. The data are the average of 2 independent experiments
and error bars represent standard deviation. Significant differences were determined using a two-way ANOVA with
Turkey's post-hoc analysis. Adjusted p-values **** p<0.0001 or ns, not significant are indicated. **B.** Comparison of
relative wall teichoic acid (WTA) levels in JE2, *pgl*, *pgl*comp, *vraG*, *pgl/vraG* and *tagO* (negative control) grown in
CDMG and CDMG OX 0.05 mg/ml. Purified WTA samples were separated on 20% native polyacrylamide (PAA) gels
before being stained with 0.1% Alcian blue. A representative image from 3 independent biological repeats is shown. **C.**
Comparison of relative lipoteichoic acid (LTA) levels in JE2, *pgl*, *pgl*comp, *pgl*::Km$^r$, *vraG*, *pgl/vraG*, USA300 LAC* and
*ltaS/gdpP* (negative control) grown in CDMG and CDMG OX 0.05 mg/ml (the *ltaS/gdpP* mutant is OX susceptible
and was only grown in CDMG). Extracted LTAs were separated on 15% PAA gels, transferred to a PVDF membrane
and probed with LTA antibody (1:5000), followed by HRP-conjugated protein G (1:2000) and colorimetric detection

with Opti-4CN substrate kit. Three independent experiments were performed, and a representative blot is shown. **D.** Mutation of *vraG* partially restores Congo Red resistance in the *pgl* mutant. 10-fold serial dilutions of JE2, *pgl*, *pgl*comp, *vraG* and *pgl*/*vraG* inoculated onto TSA supplemented with 0.125% Congo Red and grown for 24 h at 37˚C. This experiment was repeated three times and a representative plate is shown. **E.** Comparison of cell surface charge using a cyctochrome c binding assay in JE2, *pgl*, *vraG* and *pgl*/*vraG* grown in CDMG and CDMG OX 0.05 mg/ml. Positively charged cytochrome c binds more strongly to negatively charged cells. The data are the average of 3 independent experiments and error bars represent standard deviation Significant differences were determined using ordinary one-way ANOVA followed by Turkey's multiple comparison post-hoc test (* $p < 0.05$, **** $p < 0.0001$).

LTAs and cell lysis, and a significantly more positive cell surface charge which is VraFG/GraRS-dependent.

## Discussion

Beyond the central role of the oxidative phase of the PPP in producing reducing power and 5 carbon sugars for nucleotide and amino acid biosynthesis, its contribution to other phenotypes in bacteria remains much less understood due to the essentiality of most enzymes in the pathway. One exception to this is 6-phosphogluconlactonase (Pgl). Here, for the first time, we report a role for *pgl* in the control of MRSA β-lactam antibiotic resistance, growth, cell size and cell surface morphology. Our analysis revealed pleiotropic effects of the *pgl* mutation on (i) the PPP itself and downstream nucleotide biosynthesis, (ii) glycolysis and the TCA cycle and (iii) flux to cell wall, WTA and LTA precursors. All three of these pathways have previously been implicated in the control of MRSA β-lactam resistance providing a multifaceted explanation for the increased OX resistance phenotype of the *pgl* mutant.

Although OX MICs of the wild-type JE2 and *pgl* mutant were dependent on the culture media, the *pgl* mutant was always significantly more resistant and the most striking difference between the two strains was measured in chemically defined media with glucose (CDMG), which is the substrate for the PPP. Strikingly, the wild-type JE2 OX MIC was reduced to 1 μg/ml in CDMG, compared to 64 μg/ml in MHB 2% NaCl, whereas the *pgl* OX MIC was similar in both culture media (128–256 μg/ml). Given that the JE2 OX MIC was 4–16 μg/ml in CDM, in which growth of *S. aureus* is dependent on amino acid catabolism, it appears that glucose plays a significant role in controlling OX susceptibility in JE2 but not in the *pgl* mutant in which central metabolism is significantly perturbed. MHB 2% NaCl is the standard culture medium used to measure the susceptibility of *S. aureus* clinical isolates to oxacillin in diagnostic laboratories, and these experiments raise the question of whether CDMG may be more physiologically relevant in terms of predicting the *in vivo* effectiveness of β-lactams in patients with MRSA infections.

The *pgl* gene has previously been mutated in *E. coli* and *L. monocytogenes* [58, 62], leading to an accumulation of gluconate, which can be transported back into the cell and phosphorylated, thus potentially bypassing the Pgl-catalysed reaction in the PPP [58, 62]. However, in *S. aureus* the slower growth and OX resistance phenotypes of the *pgl* mutant were not dependent on the gluconate transport and catabolism genes *gntPK* or the presence of exogenous D-gluconate in the culture media. Thus, questions remain about the importance and regulation of the gluconate shunt in *S. aureus*.

In CDMG, *pgl* mutant cell size was significantly reduced compared to wild-type. Reduction in cell size may correlate with increased β-lactam resistance of *pgl*, as previously reported for a c-di-AMP phosphodiesterase *gdpP* mutant [41]. In addition to the previously reported OX-induced increase in cell size [31, 53, 59, 61], a dramatic cell lysis phenotype was also observed in wild-type JE2 grown in CDMG with sub-inhibitory OX (0.05 μg/ml), and not in the *pgl* mutant.

Not unexpectedly, the *pgl* mutation significantly perturbed the metabolome. However, accumulation of several individual metabolites within the PPP and glycolysis was not uniformly affected suggesting that the restoration of homeostasis required to enable growth in the absence of Pgl was complex. For example, downstream of Pgl in the PPP, levels of ribose-5-P were increased whereas sedoheptulose 7-P and nucleotide levels were decreased. The accumulation of ribose-5-P in a mutant lacking the transketolase *tkt* gene from the non-oxidative phase of the PPP was also accompanied by decreased sedoheptulose 7-P, although levels of inosine-5-monophosphate, xanthosine-5-monophosphate, and hypoxanthine were increased in the *tkt* mutant [48]. The reduction in purine (and pyrimidine) nucleotide accumulation in the *pgl* mutant is consistent with its sensitivity to sulfamethoxazole, and with previous studies showing that mutations in the purine biosynthesis and salvage pathways are accompanied by increased OX resistance [26, 53]. The metabolomics data presented here suggest that mutation of *pgl* was accompanied by a complex and intricately regulated interconversion of glycolytic and PPP intermediates to ensure maintenance of key central metabolites needed to support growth.

Analysis of PG architecture, crosslinking and overall concentration revealed no differences between the wild-type and *pgl* strains suggesting that other changes in the cell envelope are responsible for increased *pgl* OX resistance. However, reduced WGA binding and teichoic acid levels, particularly LTAs, is of particular interest. Hesser et al [76] recently proposed that the production of long and abundant LTAs in *S. aureus* promotes cell lysis, whereas abundant WTA levels have the opposite effect and limit cell lysis. Our data appear to be consistent with this hypothesis and revealed a correlation between significantly reduced LTA levels in *pgl* and reduced cell lysis under OX stress. In turn, the reduced cell lysis of *pgl* may also contribute to increased OX resistance. The reduced levels of UDP-Glucose and increased resistance to Congo Red, which targets LtaS were also consistence with the reduced LTA levels in *pgl*.

Strikingly, mutations in VraFG/GraSR reversed the increased OX resistance phenotype of *pgl* in CDMG, as well as increased VAN resistance in MHB and reduced Congo Red resistance. Meehl *et al* previously proposed that because mutation of *vraG* increased susceptibility to the structurally dissimilar vancomycin and polymyxin B in *S. aureus* strains Mu50 and COL, VraFG may play a broader role in the export of cell wall/teichoic acid precursors or modifying subunits, rather than specific antimicrobial agents [71]. D-alanylation of WTAs was also shown to be reduced in a *graRS* mutant [72], further implicating this multienzyme membrane complex in cell envelope biogenesis. In contrast to its effect on OX resistance, mutation of *vraG* did not reverse the impact of the *pgl* mutation on LTA levels or cell size suggesting that the broader metabolic consequences of the disrupted PPP on cell size are distinct from the more precise VraFG/GraRS-dependent increase in OX resistance.

The VraG-independent reduction in LTA levels in *pgl* raised questions about the impact of the altered WTA:LTA ratio on the *pgl* cell envelope and/or the post-translational modification(s) of teichoic acids. While construction of *pgl*/*tarM* and *pgl*/*tarS* double mutants excluded a role for α and β glycosylation of WTAs, respectively, the *pgl* mutant was significantly more positively charged in OX and this was partially reversed by the *vraG* mutation. Thus, despite the reduction in the levels of LTAs, these observations are consistent with increased D-alanylation of both LTAs and WTAs, and increased OX resistance in *pgl*.

GraSR was also shown to regulate the transcription of *mprF*, which codes for the LysPG flippase implicated in resistance to CAMPs and daptomycin [81–83]. Notably, a *mprF* missense mutation associated with increased cell size and daptomycin resistance was also shown to reduce MRSA OX resistance [84] raising the possibility that altered MprF activity could contribute to *pgl* phenotypes in a VraFG/GraRS-dependent manner.

The GraRS/VraFG complex shares significant amino acid sequence similarity with BceRS/ BceAB in *Bacillus subtilis*, which plays an important role in bacitracin resistance. Bacitracin targets the lipid II cycle intermediate undecaprenyl-pyrophosphate (UPP), which is believed to be flipped/transported across the membrane, potentially by the BecAB ABC transporter, during PG biosynthesis [85, 86]. Upregulation of *bceAB* expression by the BceR response regulator and changes in the conformation of BceAB appear to protect UPP from bacitracin inhibition [85]. Thus, while PG structure and crosslinking is unaffected by the *pgl* mutant, it is tempting to speculate that UPP flipping across the membrane by VraFG may be of particular importance for PG biosynthesis in the *pgl* mutant under OX stress in CDMG, which is detected by the GraRS two component system. GraRS is known to be required for *S. aureus* growth at high temperatures and under oxidative stress [87], supporting the conclusion that the *vraFG*-dependent increase in OX resistance in *pgl* is also environmentally-regulated, as evidenced by changes in OX MICs in different culture media.

Taken together, our data reveal dramatically increased OX susceptibility and lysis of wild-type JE2 in CDMG, which may be associated with relatively higher LTA levels compared to *pgl*. This vulnerability is apparently reversed by the reduced LTA levels and higher positive cell surface charge of *pgl* mutant cells, which are smaller and have a ruffled surface morphology, thicker cell walls and intact septa. The *pgl* OX resistance phenotype and increased positive surface charge is, in turn, dependent on VraFG and GraRS. However mutation of *vraG* in the *pgl* background did not restore LTA levels or result in an increase in cell size. We propose a possible model (Fig 9), in which the combined effects of reduced levels of LTAs and VraFG/GraRS-dependent DltABCD-mediated D-alanylation of WTAs and LTAs in *pgl* results in increased β-lactam resistance and prevents the extensive OX-induced lysis evident in the wild-type JE2.

## Materials and methods

### Bacterial strains and growth conditions

Bacterial strains and plasmids used in this study can be found in S2 Table. *Escherichia coli* strains were grown in lysogeny broth /Luria Bertani broth (LB) broth or agar (LBA) and *Staphylococcus aureus* strains were grown in Tryptic Soy Broth (TSB), Tryptic Soy Agar (TSA), Mueller-Hinton Broth (MHB) supplemented with 2% NaCl where indicated, Mueller-Hinton Agar (MHA) supplemented with 2% NaCl where indicated, Brain Heart Infusion (BHI) broth, LB broth, chemically defined medium (CDM) [38] or chemically defined medium supplemented with glucose (5 g/L) (CDMG). Culture media were supplemented with erythromycin (Erm) 10 μg/ml, chloramphenicol (Cm) 10 μg/ml, ampicillin (Amp) 100 μg/ml, kanamycin (Km) 90 μg/ml, oxacillin (OX) at varying concentrations as indicated.

### Genetic manipulation of *S. aureus*, complementation of NE202 (*pgl*) and construction of *pgl* double and triple mutants

To verify the increased OX resistance phenotype of NE202, phage 80α was used to transduce the *pgl*::Erm^r allele into wild-type JE2, as described previously [39, 57]. The presence of the *pgl*::Erm^r allele in NE202 and the transductant was verified by PCR amplification with primers NE202_check_F, NE202_check_R, Martn_ermF, and Martn_ermR (S3 Table).

To complement NE202, the *pgl* gene including its promoter and upstream regulatory sequences was first amplified from JE2 genomic DNA using *pgl*_Fwd and *pgl*_Rev primers (S3 Table), cloned into pDrive (Qiagen) in *E. coli* TOP10 (Invitrogen), verified by Sanger sequencing (Source Biosciences) before being sub-cloned on an *Eco*RI restriction fragment into the *E. coli-Staphylococcus* shuttle plasmid pLI50 [88] and transformed into *E. coli* HST08

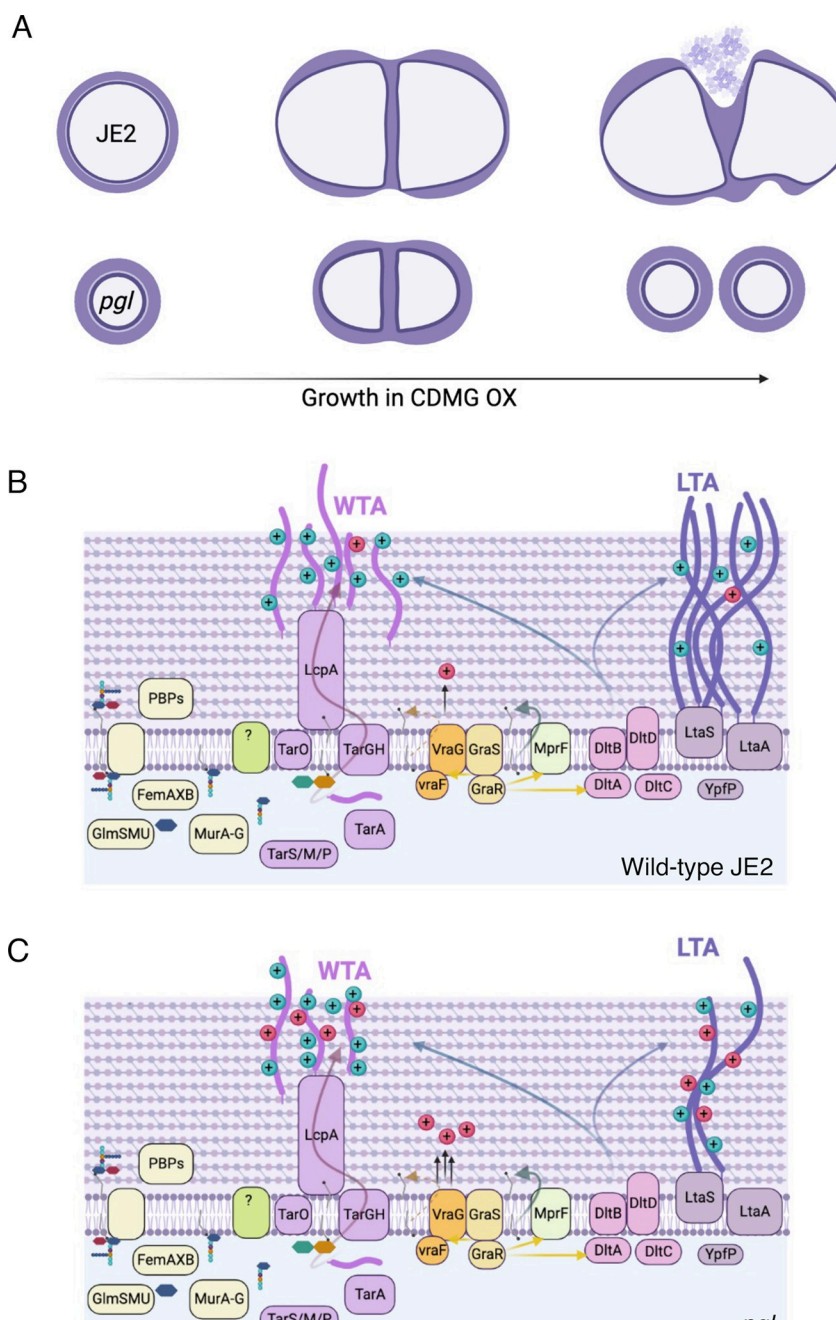

**Fig 9. Suggested model for VraFG-dependent high-level β-lactam resistance in the MRSA *pgl* mutant. A.**
Illustration of JE2 and *pgl* cell division during growth in CDMG OX. *pgl* cells are smaller than wild-type JE2 when grown in CDMG and undergo normal cell division, whereas extensive lysis is evident among wild-type cells. **B and C.**
Illustration of peptidoglycan (PG), wall teichoic acid (WTA) and lipoteichoic acid (LTA) biosynthesis in wild-type JE2 (B) and *pgl* (C). Mutations in *vraF*, *vraG* and to a lesser extent *graR* reverse the increased OX resistance phenotype of the *pgl* mutant. Metabolic reprogramming in the *pgl* mutant increases carbon flux to cell envelope precursors and β-lactam resistance via a mechanism dependent on VraFG/GraRS-controlled regulation of WTA/LTA biosynthesis, export or posttranslational modification. Previous studies have implicated the VraFG/GraRS complex in resistance to cationic antimicrobial peptides and regulation of *dltABCD* and *mprF* transcription, and it has also been proposed to play a role in the export of peptidoglycan or teichoic acid precursors or modifying subunits. Reduced levels of LTAs in *pgl* (C) compared to JE2 (B) may contribute to reduced cell lysis under OX stress. The significantly increased positive charge in the *pgl* mutant (C) compared to JE2 (B) was partially reversed by the *vraG* mutation suggesting that reduced levels of LTAs and increased VraFG/GraRS-dependent D-alanylation of WTAs and LTAs in the *pgl* mutant combine to increase OX resistance. Figure made using Biorender.com.

(Takara Bio). The pLI50_*pgl* plasmid was then isolated and transformed by electroporation into the restriction-deficient strain RN4220, and subsequently into NE202. All plasmid-harbouring strains were cultured in media supplemented with 100 μg/ml ampicillin (*E. coli*) or 10 μg/ml chloramphenicol (*S. aureus*) to maintain plasmid selection.

To generate the *pgl*::Km[r] mutant, pKAN plasmid [57] was isolated from IM08B and electroporated into NE202 (*pgl*::Erm[r]) and the Erm[r] marker swapped for the Km[r] marker using allelic exchange [57]. To construct a markerless Δ*pgl* mutant, the pTNT plasmid [57] from RN4220 pTNT was isolated and electroporated into NE202 (*pgl*::Erm[r]) and the Erm[r] marker swapped out for a shorter, markerless version of the transposon insertion leaving a small deletion in the *pgl* gene. The *pgl*::Km[r] and Δ*pgl* mutants were verified by PCR using primers NE202_check_F, NE202_check_R, KanR_fwd and KanR_rev (S3 Table)

To construct *pgl* double mutants phage 80α used to transduce the Erm[r]-marked alleles from the following Nebraska transposon library [57] mutants into *pgl*::Km[r]: NE1868 (*mecA*), NE952 (*gntP*), NE1124 (*gntK*), NE569 (*sucC*), NE547 (*sucA*), NE76 (*leuB*), NE239 (*putA*), NE1518 (*gudB*), NE70 (*vraG*), NE645 (*vraF*), NE481 (*graR*), NE942 (*tarS*), NE611 (*tarM*) NE626 (*sdhA*). The presence of transposon insertions in the genes was confirmed by PCR using primers listed in S3 Table.

To construct the *pgl*/*vraG*/*putA* triple mutant the NE239 *putA*::Erm[r] strain was first transformed with pSPC plasmid [57] isolated from IM08B pSPC and allelic exchange performed as previously described [57] to generate *putA*::Spec[r]. The *putA*::Spec[r] allele was then transduced into the *pgl*/*vraG* double mutant using phage 80α. The presence of transposon insertions in *pgl*, *vraG* and *putA* genes were confirmed by PCR using primers listed in S3 Table.

## Tecan growth curves

A Tecan Sunrise microplate instrument equipped with Magellan software was used to record data from growth experiments performed in 96-well plates. Cultures were streaked on TSA plates supplemented with antibiotics where needed and grown at 37˚C for 24 hours. The next day, colonies were resuspended in 1 ml of PBS, before being washed in PBS. The PBS washed cell suspensions were adjusted to an $OD_{600}$ of 0.2 in 1 ml of PBS and 10 μl inoculated into wells containing 190 μl growth media (MHB, LB, TSB, BHI, CDM, CDMG, CDM 10 μg/ml OX, CDMG 10 μg/ml OX, CDMG supplemented with potassium D-Gluconate (5 g/L) (with or without OX 10 μg/ml) (starting $OD_{600}$ = 0.01) and were then incubated at 35–37˚C for 24 h with shaking and $OD_{600}$ recorded every 15 min intervals. For $H_2O_2$ sensitivity assays (S5 Fig), CDMG and CDMG containing 500 μM $H_2O_2$ were inoculated at a starting $OD_{600}$ of 0.05. For human serum growth curves, human serum (from human male AB plasma, Merck) was mixed with CDMG v/v (70%-30%, 50%-50%, 25%-75%, and 10%-90% respectively). Strains were first grown at 37˚C on TSA 2% NaCl for 24 h and 5–10 colonies were resuspended in 0.85% saline before being adjusted to 0.5 McFarland standard ($A_{600}$ = 0.1). The cell suspension was then diluted 1:20 in PBS and 10 μl used to inoculate 100 μl human serum-CDMG mixture and inoculated at a starting $OD_{600}$ of 0.005. Three independent biological replicates were performed for each strain and the resulting data plotted using GraphPad Prism software V9.

## Antibiotic disc diffusion susceptibility assays

Disk diffusion susceptibility testing was performed in accordance with Clinical Laboratory Standards Institute (CLSI) guidelines [89] and as previously described [53] with the slight modifications. Briefly, overnight cultures were diluted into 5 ml fresh TSB and grown for 3 h at 37˚C with shaking at 200 rpm. The 3 h grown cultures were then adjusted to $A_{600}$ = 0.5 and 150 μl of this suspension was swabbed evenly 3 times across the surface of an MHA plate (4

mm agar thickness). Six mm blank discs (OXOID) were spotted with 20 μl antibiotics (cefoxitin 1.5 mg/ml stock). Once dried, the discs were applied onto the MHA plates spread with culture suspension before incubation for times specified by CLSI guidelines for stated antibiotics at 37˚C. Three independent measurements were performed for each strain and zone of inhibition was measured and recorded.

## Antibiotic minimum inhibitory concentration (MIC) measurements and synergy/checkerboard assays

MIC measurements by broth microdilutions were performed in accordance with CLSI methods for dilution susceptibility testing of staphylococci [90] with modifications. Briefly, strains were first grown at 37˚C on MHA 2% NaCl for 24 h and 5–10 colonies were resuspended in 0.85% saline before being adjusted to 0.5 McFarland standard ($A_{600}$ = 0.1). The cell suspension was then diluted 1:20 in PBS and 10 μl used to inoculate 100 μl media (MHB/ MHB 2% NaCl / CDM / CDMG) containing serially diluted antibiotics (oxacillin, fosfomycin, targocil, tunicamycin, Congo Red, amsacrine, DCS, vancomycin, sulfamethoxazole and polymyxin B) in 96-well plates. For human serum MICs, Human serum (from human male AB plasma, Merck) was mixed with CDMG or MHB 2% NaCl v/v (70%-30%, 50%-50%, 25%-75%, and 10%-90%) and inoculated as described above. The 96-well plates were incubated at 35˚C for 24 h and MIC values were recorded as the lowest antibiotic concentration where no growth was observed. Checkerboard/synergy assays were performed as previously described, using (0–128 μg/ml) fosfomycin and (0–256 μg/ml) oxacillin as indicated in S9 Fig.

## Genomic DNA (gDNA) extraction and Whole Genome Sequencing (WGS)

Genomic DNA (gDNA) extractions were performed using the Wizard Genomic DNA Purification Kit (Promega) following pre-treatment of *S. aureus* cells with 10 μg/ml lysostaphin (Ambi Products LLC) at 37˚C for 30 min. The genome sequencing for NE202 (*pgl*) was performed by MicrobesNG using an Illumina HiSeq platform and a 250-bp paired end read kit. DNA libraries for *pgl*::Km$^r$ and *pgl*R1 were prepared using an Illumina Nextera XT DNA Library Prep kit, validating size distribution by gel electrophoresis, and bead-normalizing the libraries. An Illumina MiSeq v2 600 cycle kit was used for genome sequencing, generating 300-bp paired end reads. PhiX was used as a sequencer loading control. The CLC Genomics Workbench software (Qiagen Version 20) was used for genome sequencing analysis of the different strains, as described previously [91]. As a reference genome, a contig was produced for wild-type JE2 by mapping Illumina reads onto the closely related USA300 FPR3757 genome sequence (RefSeq accession number NC_007793.1). The Illumina short read sequences from NTML mutants [57] of interest were then mapped onto the assembled JE2 sequence, and the presence of the transposon insertion confirmed. Single Nucleotide Polymorphisms (SNPs), deletions or insertions were identified where present.

## PBP2a Western blot analysis

PBP2a Western blots were performed as previously described [92] with slight modifications. Briefly, single colonies from wild-type JE2, *pgl* and *pgl*$_{comp}$, MSSA strain 8325–4 (negative control) and HoR MRSA strain BH1CC (positive control) were inoculated in TSB overnight and grown at 37˚C with 200 rpm shaking. The next day, day cultures were started at $OD_{600}$ 0.05 in 50 ml TSB supplemented with 0.5 μg/ml OX except for 8325–4 which was grown with no OX supplementation, and BHICC with 50 or 75 μg/ml OX, and grown for 6 hours, with shaking (200 rpm). For MHB 2% NaCl grown cells, single colonies from wild-type JE2, and *pgl* were inoculated in MHB overnight and grown at 37˚C with 200 rpm shaking. The next day, day

cultures were started at $OD_{600}$ 0.05 in 50 ml MHB 2% NaCl w/wo 32 µg/ml OX and grown for 6 hours with shaking (200 rpm). Samples were pelleted and resuspended in PBS to an $A_{600}$ = 10. Six µl of lysostaphin (10 µg/ml) and 1 µl of DNase (10 µg/ml) was added to 500 µl of this concentrated cell suspension before being incubated at 37˚C for 40 min. Next, 50 µl of 10% SDS was added, and the incubation continued for a further 20 min. The lysed cells were then pelleted in a microcentrifuge for 15 min, following which the protein-containing supernatant was collected and total protein concentration determined using the Pierce BCA Protein Assay Kit. For each sample, 8 µg total protein was run on a 7.5% Tris-Glycine gel, transferred to a PVDF membrane, and probed with anti-PBP2a (1:1000), followed by HRP-conjugated protein G (1:2000) and colorimetric detection with Opti-4CN Substrate kit. Three independent experiments were performed, and a representative image is shown.

## Peptidoglycan (PG) analysis

Wild-type JE2, *pgl* and *pgl*comp were grown in MHB and MHB supplemented with oxacillin 0.5 µg/ml, CDMG and CDMG supplemented with OX 0.05 µg/ml. For each strain and growth condition tested, independent quadruplicate 50 ml cultures were grown in flasks at 37˚C with 200 rpm shaking overnight and cell pellets were collected at 4˚C at 7000 rpm. The pellets were then resuspended in PBS, pelleted at 10000 rpm and snap frozen in liquid nitrogen in 1.5 ml tubes. Peptidoglycan (PG) was extracted from wild-type JE2, *pgl* and *pgl*comp from boiled samples as described previously [93]. Once boiled, cell wall material was pelleted by ultracentrifugation and washed with water. Clean sacculi were digested with muramidase (100 µg/ml) and soluble muropeptides reduced using 0.5 M sodium borate pH 9.5 and 10 mg/mL sodium borohydride. The pH of the samples was then adjusted to 3.5 with phosphoric acid. UPLC analyses were performed on a Waters-UPLC system equipped with an ACQUITY UPLC BEH C18 Column, 130Å, 1.7 µm, 2.1 mm × 150 mm (Waters Corporation, USA) and identified at Abs. 204 nm. Muropeptides were separated using a linear gradient from buffer A (0.1% formic acid in water) to buffer B (0.1% formic acid in acetonitrile). Identification of individual peaks was assigned by comparison of the retention times and profiles to validated chromatograms. The relative amount of each muropeptide was calculated by dividing the peak area of a muropeptide by the total area of the chromatogram. The abundance of PG (total PG) was assessed by normalizing the total area of the chromatogram to the $OD_{600}$. The degree of cross-linking was calculated as described previously [94]. The data for at least three independent experiments were plotted using GraphPad Prism software V9.

## Confocal microscopy and cell size determination

For microscopy experiments, JE2, *pgl* and *pgl*comp were grown overnight at 37˚C in CDMG w/wo 0.05 µg/ml OX. The next day, the cultures were washed and normalized to an $OD_{600}$ of 1 in PBS and 75 µl of these cultures were double stained for 30 mins at 37˚C with vancomycin-BODIPY FL at a final concentration of 2 µg/ml and WGA Alexa Fluor 594 at a final concentration of 25 µg/ml. Bacteria were then collected by centrifugation for 2 mins at 14,000 x*g*. The cells were resuspended with 100 µl of PBS, pH 7.4, and 5 µl of this sample was spotted onto a thin 1.5% agarose gel patch prepared in PBS. Stained bacteria were then imaged at X1000 magnification using Olympus LS FLUOVIEW Fv3000 Confocal Laser Scanning Microscope. Cell size was measured as previously described [54] using ImageJ software (Fiji v.1.0). Images of cells from three/four biological replicates were acquired, 50 cells measured per biological replicate (150–200 cells in total per condition), and the average and standard deviations for the three/four biological replicates were plotted using GraphPad Prism version 9.2 and significant

differences were determined using a Kruskal-Wallis test followed by a Dunn's multiple comparison test. Only 60 cells could be measured for OX treated cells due to cell lysis.

## Transmission Electron Microscopy (TEM) and cell morphology analysis

Overnight cultures of JE2, *pgl* and *pgl*$_{comp}$ were grown overnight in 5 ml CDMG at 37˚C shaking at 200 rpm. The next day, $OD_{600}$ values were measured, and cultures were used to inoculate 5 ml day cultures in CDMG 1 µg/ml OX to $OD_{600}$ of 0.06. The day cultures were grown for 4.5 hours at 35˚C shaking at 200 rpm, before being pelleted down, and normalised to $OD_{600}$ of 1 in PBS. Cells pellets were resuspended in 0.2M sodium cacodylate buffer pH 7.2. Fixed bacteria were dehydrated, embedded in resin, and thin sectioned in the University of Galway Centre for Microscopy & Imaging. Images were acquired using Hitachi H7500 Transmission Electron Microscope. Representative cells from each strain were imaged at 30,000× and 100,000× magnification.

## Congo Red susceptibility spotting assays

*S. aureus* strains JE2, *pgl*, *pgl*$_{comp}$, *vraG* and *pgl/vraG* were streaked onto TSA plates containing appropriate antibiotics, and the plates were incubated overnight at 37˚C. The next day, overnight cultures of the strains from single colonies were grown in 5 ml TSB, at 37˚C shaking at 200 rpm. The next day, PBS washed cells were normalised to an $OD_{600}$ of 1 per ml in PBS and serial dilutions prepared from $10^{-1}$ until $10^{-8}$ in a 96-well plate. Five µl of the serially diluted cell suspensions was spotted onto TSA plates containing 0.125% Congo Red. The plates were dried in a flow hood and were incubated at 37˚C for 24 hours. Plates were visualised and photos were taken for three biological replicates. Representative image is shown in Fig 8D.

## Quantification of Wheat Germ Agglutinin (WGA) binding

Overnight cultures of *S. aureus* strains were grown in 3 ml CDMG at 37˚C shaking at 200 rpm. The next day, $OD_{600}$ values were measured, and cultures were used to inoculate 5 ml day cultures in CDMG 0.1 µg/ml OX to $OD_{600}$ of 0.06. The day cultures were grown for 4.5 hours at 35˚C shaking at 200 rpm, before being pelleted down, washed with PBS, and normalised to $OD_{600}$ of 1 in PBS. One hundred µl of this cell suspension was incubated with WGA Alexa Fluor 594 at a final concentration of 25 µg/ml for 30 minutes at 37˚C. After the incubation with the dye, the cells were pelleted at 14,000 rpm for 3 minutes, and the supernatant was used for fluorescence measurements in Polarstar plate reader (Excitation/Emission 590/617 nm). PBS containing WGA Alexa Fluor 594 at a final concentration of 25 µg/ml was used as a positive control, and PBS was used as a blank control. The reduction in WGA Alexa Fluor 594 from the positive control was calculated per sample, and % bound WGA plotted using were plotted and significant differences were determined for two biological repeats using two-way ANOVA with Turkey's post-hoc. using GraphPad Prism version 9.2

## Culture supernatant sample preparation for LC-MS/MS

Overnight cultures of *S. aureus* strains were grown in 3 ml CDMG at 37˚C shaking at 250 rpm. The next day, 250 ml flasks containing 25 ml CDMG were inoculated to an $OD_{600}$ of 0.06 and were grown for 7.5 h ($OD_{600}$ = 4.22–4.96). One ml from the cultures were collected and centrifuged at 12,000 rpm, 10 min at 4˚C, and supernatant collected. These samples were diluted 1:100 v/v using 10 mM $NH_4OAc$ + 10mM $NH_4OH$ + 5% acetonitrile. 5 µl was injected into the LC-MS/MS (details below).

## Sample preparation intracellular metabolite analysis by LC-MS/MS

Overnight cultures of *S. aureus* strains were grown in 3 ml CDMG at 37°C shaking at 250 rpm. The next day, 250 ml flasks containing 25 ml CDMG (with or without 1 µg/ml OX) were inoculated at a starting $OD_{600}$ of 0.06 and grown for 4–5 hours (until exponential phase was reached) at 37°C shaking at 250 rpm. Culture volumes corresponding to $OD_{600}$ of 10 were then harvested and rapidly filtered through a membrane (0.45 µm, Millipore). The cells on the membrane were washed twice with 5 ml cold saline and immediately quenched in ice-cold 60% ethanol containing 2 µM Br-ATP as an internal control. The cells were mechanically disrupted using a bead homogenizer set to oscillate for 3 cycles (30 s) of 6800 rpm with a 10 s pause between each cycle. Cell debris was separated by centrifugation at 12,000 rpm. The supernatant containing intracellular metabolites were lyophilized and stored at -80°C. These samples were reconstituted in 100 µl of 50% MeOH.

## Analysis of PPP flux using 1,2-$^{13}$C glucose

The *S. aureus* strains were inoculated in 25 ml CDM containing either unlabelled or 1,2-$^{13}$C-labeled glucose at a starting $OD_{600}$ of 0.06. The cultures were grown at 37°C with shaking at 250 rpm until the $OD_{600}$ reached 1.0. The culture volume corresponding to an $OD_{600}$ of 10 was then harvested and immediately filtered through a 0.45 µm Millipore membrane before being subjected to further processing as outlined in the previous section.

## LC-MS/MS mass spectrometry

A triple-quadrupole-ion trap hybrid mass spectrometer (QTRAP6500+ by Sciex, USA) connected to an ultra-performance liquid chromatography I-class (UPLC) system (Waters, USA) was utilized for metabolomics analysis. The chromatographic separation was performed using the UPLC on a XBridge Amide analytical column (150 mm x 2.1 mm ID, 3.5 µm particle size by Waters, USA) and a binary solvent system with a flow rate of 0.3 ml/min. The analytical column was preceded by a guard XBridge Amide column (20 mm x 2.1 mm ID, 3.5 µm particle size by Waters, USA). The mobile phase A consisted of 10 mM ammonium acetate and 10 mM ammonium hydroxide with 5% acetonitrile in LC-MS grade water (pH adjusted to 8.0 with glacial acetic acid), while mobile phase B was 100% LC-MS grade acetonitrile. The column was maintained at 40°C and the autosampler temperature was kept at 5°C throughout the sample run. The gradient conditions were as follows: A/B ratio of 15/85 for 0.1 minute, 16/84 for 3.0 minutes, 35/65 for 4.0 minutes, 40/60 for 5.0 minutes, 45/55 for 3.0 minutes, 50/50 for 5.5 minutes, 30/70 for 1.5 minutes, and finally equilibrated at 15/85 for 5.0 minutes before the next run. The needle was washed with 1000 µL of strong wash solvent (100% acetonitrile) and 1000 µL of weak wash solvent (10% aqueous methanol) prior to injection, with an injection volume of 5 µL. The QTRAP6500+ operated in polarity switching mode for the targeted quantitation of amino acids through the Multiple Reaction Monitoring (MRM) process. The electrospray ionization (ESI) parameters were optimized, with an electrospray ion voltage of -4200 V in negative mode and 5500V in positive mode, a source temperature of 400°C, a curtain gas of 35 and gas 1 and 2 of 40 and 40 psi, respectively. Compound-specific parameters were optimized for each compound through manual tuning, with declustering potential (DP) of 65V in positive mode and -60V in negative mode, entrance potential (EP) of 10V in positive mode and -10V in negative mode, and collision cell exit potential (CXP) of 10V in positive mode and -10V in negative mode.

## Extraction and visualisation of WTA

The WTA extraction and visualization was performed as previously described [34, 95, 96] with modifications. Briefly, wild-type JE2, *pgl*, *vraG*, *pgl/vraG*, *tagO* (LAC* Δ*tagO*, ANG4759 [97]) were grown in CDMG and CDMG supplemented with OX 0.5 μg/ml (except *tagO*) overnight. For each strain and growth condition tested, independent triplicate 50 ml cultures were grown in flasks at 35˚C with 200 rpm shaking overnight. Cell pellets (OD$_{600}$ of 50) were collected at 10˚C at 7800 rpm. The pellets were then washed with 1 ml of 50 mM MES, pH 6.5, resuspended in 4% SDS and 50 mM MES, pH 6.5 and incubated at 100˚C for 1 hr. The boiled samples were washed twice in 4% SDS and 50 mM MES, once with 2% NaCl and 50 mM MES, and once with 50 mM MES before being resuspended in 1 mL 20 mM Tris-HCl (pH 8), 0.5% SDS, and 20 μg/ml proteinase K. The samples were incubated for 4 hrs at 50˚C (shaking at 1,400 rpm in a thermomixer). The pellet was recovered, washed with 2% NaCl and 50 mM MES, and washed three times with water before being resuspended in 100 μl 0.1 M NaOH and incubated at 20˚C for 12 h (shaking at 1,400 rpm). Samples were then heated to 65˚C for 1 hr, followed by centrifugation at 14,000 rpm for 1 min. The supernatant was removed into a fresh 1.5 ml Eppendorf tube, and neutralized with 25 μL 1 M Tris-HCl, pH 8, and stored at −20˚C.

Purified WTA samples were mixed with 2 M Sucrose (1:4 ratio), and 12.5 μl loaded on 20% native polyacrylamide gels. The gels were run at 120 V in Tris-Tricine running buffer (0.1 M Tris, 0.1 M Tricine) before being stained with 0.1% Alcian blue in 3% acetic acid for 30 minutes at RT on a shaker, covered in aluminium foil. The gels were de-stained with water and a representative image from 3 independent biological repeats are shown.

## Extraction of LTA and detection by Western blot Analysis

LTA extraction and visualization was performed as previously described [34, 95, 96], with modifications. Briefly, wild-type JE2, *pgl*, *pgl*$_{comp}$, *pgl*::Km$^r$, *vraG*, *pgl/vraG*, wild-type LAC* [98] and *ltaS/gdpP* (ANG2434) [41] were grown in CDMG and CDMG supplemented with OX 0.05 μg/ml (except *ltaS/gdpP*) overnight. For each strain and growth condition tested, independent triplicate 5 ml cultures were grown in 30 ml universals at 35˚C with 200 rpm shaking overnight. Cell pellets (OD$_{600}$ of 3) were collected at 10˚C at 7800 rpm. The pellets were then resuspended in 1 ml PBS and transferred to screw cap tubes containing approximately 100 μL of 100 μm glass beads. The cells were homogenised at room temperature using a FastPrep-24 (MP Biomedicals) for 3 cycles of 6 m/s for 40 s each, followed by a 4 mins rest. The tubes were then centrifuged at 200 × g for 1 min to settle the beads, and 700 μl of the supernatant transferred into a new tube, and the cell debris were pelleted at 14,000 rpm for 15 minutes. The supernatant was removed, and the remaining pellet containing LTAs was resuspended in 80 μL 2× Laemmli sample buffer. The samples were incubated at 95˚C for 20 mins, centrifuged at 14,000 rpm for 5 mins, and the supernatant stored at −20˚C.

LTA extracts (20 μl) were run on 15% polyacrylamide gels and transferred to PVDF membranes using a Trans-Blot Turbo transfer system (Bio-Rad). After blocking with 5% milk and 1% BSA in TBST for 3 hrs, LTA was detected with an anti-LTA primary antibody (MAb 55; HycultBiotech; 1:5000 dilution), followed by HRP-conjugated protein G (1:2000) and colorimetric detection with Opti-4CN Substrate kit. Three independent experiments were performed, and a representative image is shown.

## Cytochrome c binding assay

Cytochrome c binding assay was performed as previously described [67, 99] with modifications. Briefly, wild-type JE2, *pgl*, *vraG* and *pgl/vraG* were grown in CDMG and CDMG supplemented with OX 0.5 μg/ml overnight. For each strain and growth condition tested,

independent triplicate 50 ml cultures were grown in flasks at 35˚C with 200 rpm shaking over-night. Cell pellets ($OD_{600}$ of 10) were collected by centrifugation at 7800 rpm. The pellets were then washed twice and resuspended in 20 mM MOPS buffer (pH 7.0), mixed with cytochrome c (0.5 mg/ml final concentration) and incubated at room temperature for 10 minutes. The cells were then centrifuged, and the amount (% bound) of cytochrome c left in the supernatant was determined by measuring the $OD_{410}$ and comparing it to the relative (100%) cytochrome c control (0.5 mg/ml).

## Supporting information

**S1 Table. Oxacillin minimum inhibitory concentrations (MICs, µg/ml) of JE2 and *pgl* grown in CDMG or MHB 2% NaCl supplemented with 10–70% (v/v) human serum.** (DOCX)

**S2 Table. Bacterial strains and plasmids used in this study.** (DOCX)

**S3 Table. Oligonucleotide primers used in this study.** (DOCX)

**S1 Fig. Comparison of PBP2a expression in wild-type JE2 and *pgl* by Western blot. A.** JE2, *pgl* (NE202), *pgl*::Km[r], and *pgl~comp~* grown for 6 h in TSB supplemented with OX 0.5 mg/ml. HoR MRSA strain BH1CC (positive control) was grown in TSB OX 50 and OX 75 mg/ml, and MSSA strain 8325–4 (negative control) was grown in TSB alone. **(B)** JE2 and *pgl* grown for 6 h in MHB 2% NaCl alone or supplemented with OX 32 mg/ml. For each sample, 8 µg total protein was run on a 7.5% Tris-Glycine gel, transferred to a PVDF membrane and probed with anti-PBP2a (1:1000), followed by HRP-conjugated protein G (1:2000) and colorimetric detection with Opti-4CN Substrate kit. Three independent experiments were performed, and representative blots are shown. (TIF)

**S2 Fig. Impact of the *pgl* mutation on growth in different culture media. A.** Isolated colonies of JE2 (left) and *pgl* (right) after growth on MHA for 24 h at 37˚C. **B-G.** Growth of JE2, *pgl* and the complemented *pgl* mutant for 25 hrs at 37˚C in Mueller Hinton broth, MHB (B), Luria Bertani, LB (C), Tryptic Soya broth, TSB (D), Brain Heart Infusion, BHI (E), Chemically defined media with glucose, CDMG (F) and chemically defined media with no glucose, CDM (G). Growth ($OD_{600}$) was measured at 15 min intervals in a Tecan plate reader. Data are the average of 3 independent experiments plotted using GraphPad Prism V9 and error bars represent standard deviation. (TIFF)

**S3 Fig. Oxacillin (OX) resistance in JE2 is dependent on physiologically relevant glucose concentrations. A.** Growth of wild-type JE2 and *pgl* for 24 hrs at 37˚C in chemically defined media (CDM) supplemented with 0.07 g/l (0.43 mM) glucose alone or with OX 10 mg/ml. Growth ($OD_{600}$) was measured at 15 min intervals in a Tecan plate reader. Data are the average of 3 independent experiments plotted using GraphPad Prism V9 and error bars represent standard deviation. **B.** Oxacillin MICs of JE2 and *pgl* in CDM with no glucose (CDM), or CDM supplemented with glucose concentrations from 0.03–5 g/l (1.75–28 mM). Note that the Y axis (Oxacillin MIC) is a log2 scale. (TIF)

**S4 Fig. The *pgl* mutant is more resistant to oxacillin (OX) than wild-type JE2 when grown in human serum. A and B.** Growth of JE2 (A) and *pgl* (B) for 24 hrs at 35˚C in CMDG 25% human serum supplemented with OX 1, 32 and 256 mg/ml. **C and D.** Growth of JE2 (C) and *pgl* (D) for 24 hrs at 35˚C in CMDG 70% human serum supplemented with OX 1, 32 and 256 mg/ml. Growth ($OD_{600}$) was measured at 15 min intervals in a Tecan plate reader. Data are the average of 3 independent experiments plotted using GraphPad Prism V9 and error bars represent standard deviation.
(TIF)

**S5 Fig. Mutation of *pgl* increases sensitivity to oxidative stress.** Growth of wild-type JE2 and *pgl*::Km[r] for 24 hrs at 37˚C in CDMG or CDMG supplemented with 500 mM $H_2O_2$. Growth ($OD_{600}$) was measured at 15 min intervals in a Tecan plate reader. Data are the average of 3 independent experiments plotted using GraphPad Prism V9 and error bars represent standard deviation.
(TIFF)

**S6 Fig. Comparison of peptidoglycan oligomerisation, cross-linking and total amount in JE2, *pgl* and the complemented *pgl* mutant (*pgl*comp).** A-D. Relative proportions of cell wall muropeptide fractions based on oligomerization relative cross-linking efficiency in peptidoglycan extracted from JE2, *pgl* and *pgl*_comp grown to exponential phase in CDMG (A), CDMG supplemented with OX 0.05 μg/ml (B), MHB (C) and MHB supplemented with OX 0.5 mg/ml (D). E-H. Total peptidoglycan (PG) extracted from normalised cell extracts of JE2, *pgl* and *pgl*_comp grown to exponential phase in CDMG (E), CDMG supplemented with OX 0.05 μg/ml (F), MHB (G) and MHB supplemented with OX 0.5 mg/ml (H). The total PG content was calculated as the area below the chromatogram peaks/$OD_{600}$ and mean and standard deviation from three/four biological repeats plotted using GraphPad Prism V9.
(TIFF)

**S7 Fig. Exogenous addition of D-gluconate or mutation of the gluconate shunt genes *gntP* or *gntK* did not restore wild-type oxacillin (OX) resistance in the *pgl* mutant. A and B.** Growth of JE2, *pgl* and the complemented *pgl* mutant for 25 hrs at 35˚C in CDMG supplemented with 5g/l potassium D-gluconate (0.5%) and no OX (A) or with both D-gluconate (0.5%) and OX 10 mg/ml (B). **C.** Growth of JE2, *pgl*, *pgl*_comp *pgl*::Km[r], *gntP* (NE952), *gntK* (NE1124), *pgl*/*gntP* and *pgl*/*gntK* for 25 hrs at 35˚C in CDMG supplemented with OX 10 mg/ml. Growth ($OD_{600}$) was measured at 15 min intervals in a Tecan plate reader. Data are the average of 3 independent experiments using GraphPad Prism V9 and error bars represent standard deviation.
(TIFF)

**S8 Fig. Mutations in amino acid and TCA cycle biosynthetic genes did not reverse increased OX resistance in the *pgl* mutant. A and B.** Comparison of amino acid (A) and TCA cycle metabolites (B) in supernatants of JE2, *pgl* and *pgl*_comp cultures grown for 7.5 h in CDMG measured by HPLC. Cell densities ($OD_{600}$) were normalized to each other before the cells were pelleted and supernatants collected. The data (CPS) shown are the average of three independent experiments and standard deviations are shown[‡]. Statistical significance was determined using a 2-way Anova with Dunnett's multiple comparison test; * $p < 0.05$, ** $p < 0.01$, ***$p < 0.001$, ****$p < 0.0001$. **C and D.** Growth of JE2, *pgl*, *pgl*::Km[r], *pgl*_comp, *putA*, *gudB*, *sdhA*, *sucA*, *sucC*, *pgl*/*putA*, *pgl*/*gudB*, *pgl*/*sdhA*, *pgl*/*sucA* and *pgl*/*sucC* for 25 hrs at 35˚C in CDMG (C) and CDMG supplemented with OX 10 mg/ml (D). Growth ($OD_{600}$) was measured at 15 min intervals in a Tecan plate reader. Data are the average of 3 independent

experiments using GraphPad Prism V9 and error bars represent standard deviation.
(TIF)

**S9 Fig. Mutation of *pgl* significantly increases resistance to a combination of oxacillin and fosfomycin.** Checkerboard titration assays were conducted using fosfomycin and oxacillin with (A) JE2 and (B) *pgl*, grown for 24 h in Mueller Hinton 2% NaCl broth in 96-well plates. The data shown are the $OD_{600}$ values for each well. The experiments were repeated at least three times and the data from a representative 96-well plate is shown. Green shaded boxes indicated wells in which significant growth was measured.
(TIFF)

**S10 Fig.** Comparison of wild-type JE2 (A), *pgl* (B) and *pgl*comp (C) cells grown in CDMG without oxacillin (OX) using transmission electron microscopy at 30,000× (left) and 100,000× (right) magnification. Cells were collected from exponential phase cultures grown for 4.5 h in CDMG normalized to $OD_{600} = 1$ in PBS before being fixed and thin sections prepared. Representative cells from each strain are shown. Scale bars represent 500 nm at 30,000× or 100 nm at 100,000× magnification.
(TIFF)

**S11 Fig. Mutations in *vraF*, *vraG* and *graR* reverse the increased oxacillin (OX) minimum inhibitory concentration (MIC) of *pgl* mutants in CDMG.** OX MICs (mg/ml) of JE2, *pgl*, *pgl*comp *pgl*::Km$^r$, *vraG*, *vraF*, *putA*, *graR*, *tarM*, *tarS*, *pgl*/*tarS*, *pgl*/*tarM*, *pgl*/*vraG*, *pgl*/*vraF*, *pgl*/*putA*, *pgl*/*graR*, *pgl*/*putA*/*vraG* and *pgl*R1 were measured by the broth microdilution method in CDMG. The MIC was measured in two independent experiments for each strain and variation plotted using GraphPad Prism V9 shown. Note that the Y axis (Oxacillin MIC) is a log2 scale.
(TIFF)

**S1 Data. Numerical data used to generate graphs and histograms (Fig 2A–2C, S2B–S2G, 3C, S3A–S3B, S4A–S4D, S5, S6A–S6H, S7B–S7C, S8A–S8B, 4B–4C, S8C–S8D, 5, 7A, 7B, 7D, 8A, 8E, S11).**
(ZIP)

## Acknowledgments

We are grateful to Dr Peter Owens and Dr Emma McDermott from the University of Galway Centre for Microscopy & Imaging (https://imaging.universityofgalway.ie) for their technical and scientific assistance with confocal and electron microscopy analysis, and to Volkhard Kaever, Hannover Medical School for preliminary metabolomic analysis.

M.S.Z. and J.P.O'G. conceptualized the study. Formal analysis was performed by M.S.Z., L.A.G., E.B., A.C.N., J.A., E.O'N., F.R., P.D.F., F.C., V.C.T and J.P.O'G. The investigation and methodology were performed by M.S.Z., L.A.G., E.B., A.C.N., J.A., D.S., M.S., Ó.B. and F.R. The data was curated by M.S.Z. The original draft of the manuscript was written by M.S.Z. and J.P.O'G. and was reviewed, edited, and approved by all authors. Funding was acquired by E. O'N., P.D.F., F.C., V.C.T. Ó.B. and J.P.O'G. The project was administered by J.P.O'G.

## Author Contributions

**Conceptualization:** Merve S. Zeden, James P. O'Gara.

**Data curation:** Merve S. Zeden.

**Formal analysis:** Merve S. Zeden, Laura A. Gallagher, Emilio Bueno, Aaron C. Nolan, Jongsam Ahn, Fareha Razvi, Paul D. Fey, Felipe Cava, Vinai C. Thomas, James P. O'Gara.

**Funding acquisition:** Eoghan O'Neill, Paul D. Fey, Felipe Cava, Vinai C. Thomas, James P. O'Gara.

**Investigation:** Merve S. Zeden, Laura A. Gallagher, Emilio Bueno, Aaron C. Nolan, Jongsam Ahn, Dhananjay Shinde, Margaret Sladek.

**Methodology:** Merve S. Zeden, Laura A. Gallagher, Emilio Bueno, Aaron C. Nolan, Jongsam Ahn, Dhananjay Shinde, Margaret Sladek, Órla Burke.

**Project administration:** James P. O'Gara.

**Software:** Merve S. Zeden.

**Supervision:** Merve S. Zeden, Paul D. Fey, Felipe Cava, Vinai C. Thomas, James P. O'Gara.

**Validation:** Merve S. Zeden.

**Visualization:** Merve S. Zeden.

**Writing – original draft:** Merve S. Zeden, Laura A. Gallagher, James P. O'Gara.

**Writing – review & editing:** Merve S. Zeden, Laura A. Gallagher, Emilio Bueno, Aaron C. Nolan, Jongsam Ahn, Dhananjay Shinde, Fareha Razvi, Margaret Sladek, Órla Burke, Eoghan O'Neill, Paul D. Fey, Felipe Cava, Vinai C. Thomas, James P. O'Gara.

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
