## [Decision Letter · Decision Letter 0]

11 Apr 2023

Dear Dr Zeden,

Thank you very much for submitting your manuscript "Metabolic reprogramming and flux to cell envelope precursors in a pentose phosphate pathway mutant increases MRSA resistance to β-lactam antibiotics" for consideration at PLOS Pathogens. As with all papers reviewed by the journal, your manuscript was reviewed by members of the editorial board and by several independent reviewers. In light of the reviews (below this email), we would like to invite the resubmission of a significantly-revised version that takes into account the reviewers' comments.  Specifically, one of the reviewers was unsure of the relevance for Staph aureus as a pathogen - experiments that show that the metabolic reprogramming is relevant for infection and disease, in vivo or in a model of infection, can help solidify the importance of this process beyond laboratory conditions. 

We cannot make any decision about publication until we have seen the revised manuscript and your response to the reviewers' comments. Your revised manuscript is also likely to be sent to reviewers for further evaluation.

Sincerely,

Gongyi Zhang

Academic Editor

PLOS Pathogens

Marcel Behr

Section Editor

PLOS Pathogens

Kasturi Haldar

Editor-in-Chief

PLOS Pathogens

orcid.org/0000-0001-5065-158X

Michael Malim

Editor-in-Chief

PLOS Pathogens

orcid.org/0000-0002-7699-2064

Reviewer's Responses to Questions

**Part I - Summary**

Reviewer #1: In their manuscript entitled “Metabolic reprogramming and flux to cell envelope precursors in a pentose phosphate pathway mutant increases MRSA resistance to β-lactam antibiotics”, Zeden et al investigated the metabolic and cell wall composition changes in an MRSA mutant lacking the 6-phosphogluconolactonase gene pgl that contribute to increased resistance to β-lactams. Pgl is the second enzyme in the oxidate phase of the Pentose Phosphate Pathway, and is, interestingly, the only non-essential gene in this pathway. The authors show that the pgl mutant has a higher MIC than WT for oxacillin (OX), as well as other β-lactams and bactericidal compounds that target the cell membrane. This phenotype is more pronounced in chemically defined media with glucose (CDMG) and absent in media without glucose (CDM), suggesting that carbon metabolism may have a role in the mechanism of resistance. Interestingly, the pgl mutant grows more slowly and has a reduced cell size compared to WT. Metabolomic analysis of bacteria grown in CDMG with or without OX revealed changes in key intermediates, suggesting that carbon flux was redirected from cell wall precursors. The authors hypothesize that changes to composition or architecture of the cell wall may be responsible for the antibiotic resistance phenotype. However, although the cell surface morphology is different, and the cell wall and septa between dividing cells is thicker in the mutant, the authors did not find changes to the peptidoglycan amount, cross-linking or oligomerization. The authors identify a mutant in the efflux pump VraG that reverses the resistance phenotype of the pgl mutant. Interestingly the double mutant still has slower growth kinetics and a smaller cell size. Wall teichoic acids have been shown to be required for b-lactam resistance in MRSA. Lower wheat germ agglutinin staining of the pgl mutant, compared to the WT or the double pgl/vraG mutant, showed lower levels of teichoic acids on the membrane.

This is a well written paper that includes helpful diagrams in the figures to orient the reader to the topic and specific experiments. The data is presented in a logical and easy to follow manner. Although the authors were unable to pinpoint the specific metabolic changes or the changes to the peptidoglycan that may be responsible for the increased OX resistance of the pgl mutant, the connections between the PPP and the VraFG/GraRS pathways is nevertheless novel and interesting.

Reviewer #2: The manuscript by Zeden et al. is an interesting study that seeks to understand the mechanistic underpinnings of high-level oxacillin resistance in a strain of Staphylococcus aureus lacking 6-phosphogluconolactonase, the second enzyme in the oxidative branch of the pentose phosphate pathway (PPP). S. aureus is a major human pathogen that is responsible for skin and soft tissue infections and devastating deep-seated infections. High level b-lactam resistance compounds the problem and thus, understanding the mechanisms of underlying resistance can help identify new therapeutic approaches and drug targets. The authors make the mutant (pgl) and characterize its phenotypes, noting that resistance is glucose dependent, and dependent on the alternative transpeptidase MecA (PBP2a). They rule out changes to peptidoglycan structure, TCA cycle, and amino acid metabolism, and provide evidence supporting redirected flux to cell wall precursors contributes to the resistance phenotype. Additional cell envelope changes appear to result from altered activities of teichoic acid biosynthetic enzymes. Inactivating the GraRS two-component system and VraFG ABC efflux pump re-sensitizes the pgl mutant to the b-lactam oxacillin.

The study is interesting, and the experiments are methodical in nature to unravel the mechanism. There is a great deal of attention to detail.

Reviewer #3: This manuscript by Zeden et al explores the role of the pentose phosphate pathway, specifically examining phenotypes associated with mutation of the pgl gene, in resistance to b-lactam antibiotics in MRSA.

This manuscript is well written and outlines a logical series of experiments carried out based on antibiotic MICs, and observations of cell growth kinetics and hypotheses based upon knowledge of microbial/staphylococcal metabolism.

**Part II – Major Issues: Key Experiments Required for Acceptance**

Reviewer #1: 1. The western blot in Figure SF1 led the authors to conclude that there are no differences in the levels of PBP2a that would explain the increased resistance of the pgl mutant. This data should be further validated. This experiment was done in TSB + OX at 6hr, when there is a slight growth difference (Fig S2D); additionally, there is no data presented about the susceptibility of the pgl strain under these conditions: if there are no differences in susceptibility in TSB then a negative result may not be informative of what is happening in CDMG. A more appropriate condition would be in CDMG OX10 at an early time point before the WT and pgl growth/survival curves diverge. Quantification of the western blot should also be presented to further validate that there is no difference in PBP2a protein expression.

2. The authors conclude that metabolic reprograming is at least partially responsible for the increased OX resistance of the pgl mutant. The experiments leading to this conclusion were done in CDMG comparing WT to the pgl mutant, where the difference in OX resistance is evident between WT and pgl. Are any of these changes also event in CDM, where both strains are susceptible to OX? If so, wouldn’t that imply that these changes are not responsible for the increased resistance? In order to conclude that the metabolic reprograming is correlated to resistance, it would be important to show how metabolism is affected under conditions where the pgl mutant is susceptible to OX.

3. Are levels of VraFG higher in the pgl mutant in CDMG? A reasonable hypothesis would be that in the presence of exogenous glucose the pgl mutant has higher expression of this efflux pump. Confirming the level of VraF and VraG also in the absence of exogenous glucose may also help explain the resistance phenotype of the pgl/vraGF mutants grown in MHb 2% NaCl.

Reviewer #2: There are multiple major issues to address. First, there are so many details and so much information that the manuscript is dense and difficult to digest. The model at the end does not articulate a clear explanation for the resistance phenotype. Synthesizing a better model would help. Second, it is not at all clear the role of the Gra-Vra complex in increased b-lactam resistance. The suppressor mutant was isolated during maker swap, so the selective pressure is unclear. Further, it is not clear how a mutation upstream of the genes results in the same phenotype as a transposon mutant. Third, the Gra-Vra experiments, while interesting, don’t add to the overall model. What’s the linkage between this membrane complex and the changes to the envelope? Is the activity of Gra higher in the pgl mutant? How might this happen? It seems as though this part of the story is incomplete and might be suited for a separate manuscript.

Reviewer #3: The role of glucose in the phenotypes observed is clear. The authors seem to have only used one concentration of glucose in their media that, in the presence of OX, induces increased cell size and lysis. I should think it informative to perform titration experiments, varying the concentration of glucose in the media, to follow the phenotypes, including growth kinetics, cell size, antibiotic resistance. This would be important given the glucose concentration (5g/L or 28 mM) employed here far exceeds physiological concentrations.

With the use of the carefully controlled experiments, along with the various mutant phenotypes, the work does add significantly more information to the role of metabolic flux in changing sensitivity to cell wall active antibiotics. Although a mechanism was not fully elucidated, the work suggests that increased carbon flux from glucose to cell envelope precursors is responsible for the observed changes in antibiotic resistance. It would add significant strength to this assumption were the exact changes in the cell envelope to have been determined. The data do imply that the impact of the pgl mutation on WTA and/or antibiotic MIC is only evident under high glucose settings. The bacteria make WTA even in TSB so experiments to more clearly identify structural alterations in WTA or LTA as a function of pgl mutation and glucose would add support to the model.

The experiments presented here show that under some culture conditions glucose and pgl mutation can affect antibiotic MIC and cell ultrastructure. Of interest to the readers of PLOS Pathogens would be to know how these changes affect in vivo pathogenesis. Even using a relevant in vivo substitute in in vitro experiments, like growth phenotypes/MICs in human serum (where the glucose concentration is lower) would be informative. Are there any in vivo phenotypes associated with the pgl mutants, in MRSA or MSSA, with or without the presence of antibiotic?

**Part III – Minor Issues: Editorial and Data Presentation Modifications**

Reviewer #1: -Lines 263/264 are confusing: if I understand correctly, the “growth… was similar” statement refers to growth with or without gluconate; as written, the sentence implies that the growth was similar between gluconate alone and OX.

-The resistance phenotype of pgl/gntP and pgl/gntK double mutants is taken as evidence that the gluconate shunt genes do not play a role in the pgl OX resistance phenotype. However, in Fig S5C, the double mutants do show a growth delay in the presence of OX, rather than phenocopying the pgl mutant alone. This suggests that the gluconate shunt genes have a partial role, so the conclusion of this experiment, as stated, is too strong.

-Statistics should be added to Fig S6A/B since the differences between the conditions are difficult to judge by eye. For panel A/threonine and panel B/a-KG, where there no measurements for the JE2 and pglcomp strains or are these values 0?

-Lines 417/418 refer to the growth of the pgl/graR mutant in CDMG OX in Figure 7B. However, this panel does not show that particular mutant.

-I could not find the time point of the metabolomic analysis presented in Figure 5 either in the figure legend or in the materials and methods section. Please include this information.

-Can the authors comment in the discussion on what the potential mechanism for decreased cell size and increased cell wall thickness may be? Given that the pgl/vraG mutant does not reverse that phenotype, but does reverse the resistance phenotype, one would conclude that these changes to the cell wall are not in themselves sufficient for β-lactam resistance. Can the authors comment on the relationship between these two phenotypes?

-For Figure 9, panel B: it would be helpful to lay out the model contrasting WT vs pgl in two separate diagrams. As drawn, it’s hard to follow what the authors propose is happening in the pgl mutant that is different from the WT strain.

-It would be great if the authors could highlight in more depth the glucose-dependent antibiotic resistance of the pgl mutant and its relationship to clinical antibiotic testing. The authors touch on this in the discussion (lines 511-515), but given the significant implications of this, another sentence or two would be beneficial.

Reviewer #2: Additional minor issues include:

1. Line 210: do you mean glucose-dependent? Please rephrase

2. Line 225: it’s significant or not. Do you mean more pronounced or exaggerated? Please rephrase

3. Line 307: use of the leuB mutant needs better justification.

4. Fig 7: It’s hard to see the strains and evaluate the data. Perhaps a bar graph with final OD values would be better here and in similar figures.

5. Line 409: please delete the repeated reference to the figure

6. Lines 456-458: Is this really restored to WT levels? Statistical analysis can help here.

7. Please provide a reference for the CDM formulation

8. LB is lysogeny broth, not Luria Broth. See reference:

9. A better description of how the cells were grown is required to reproduce the experiments elsewhere, especially since the extent of aeration can affect metabolism. With shaking? What was the vessel-medium ratio?

10. References beyond 92 are missing, yet are cited in the supplementary table (for instance ref 96)

Reviewer #3: y-axis on fig 2A is not scaled properly.

Fig9A can be made more clear? For instance, what is shown in the division septum in the top right? Why would it not be present in the pgl mutant depicted below?

PLOS authors have the option to publish the peer review history of their article (what does this mean?). If published, this will include your full peer review and any attached files.

Reviewer #1: No

Reviewer #2: No

Reviewer #3: No
---

## [Decision Letter · Decision Letter 1]

4 Jul 2023

Dear Dr Zeden,

We are pleased to inform you that your manuscript 'Metabolic reprogramming and altered cell envelope characteristics in a pentose phosphate pathway mutant increases MRSA resistance to β-lactam antibiotics' has been provisionally accepted for publication in PLOS Pathogens.

Best regards,

Gongyi Zhang

Academic Editor

PLOS Pathogens

Marcel Behr

Section Editor

PLOS Pathogens

Kasturi Haldar

Editor-in-Chief

PLOS Pathogens

orcid.org/0000-0001-5065-158X

Michael Malim

Editor-in-Chief

PLOS Pathogens

orcid.org/0000-0002-7699-2064

Reviewer Comments (if any, and for reference):

Reviewer's Responses to Questions

**Part I - Summary**

Reviewer #1: In the resubmission of their manuscript, Zeden et al have provided additional data and analysis that strengthen their conclusions. Additions and edits of the text and model diagrams are helpful. Although they did not perform some of the experiments requested by the reviewers, their justifications are satisfactory.

Reviewer #2: The authors have done a nice job addressing my previous comments. I have no further concerns.

Reviewer #3: A well executed study

**Part II – Major Issues: Key Experiments Required for Acceptance**

Reviewer #1: (No Response)

Reviewer #2: none

Reviewer #3: The authors have added significant new experimental data, included that which addresses comments I raised in the first round of review. I am satisfied with their responses to the concerns raised.

**Part III – Minor Issues: Editorial and Data Presentation Modifications**

Reviewer #1: (No Response)

Reviewer #2: none

Reviewer #3: Minor issues were addressed

PLOS authors have the option to publish the peer review history of their article (what does this mean?). If published, this will include your full peer review and any attached files.

Reviewer #1: No

Reviewer #2: No

Reviewer #3: No

---

## [Editor Report · Acceptance letter]

19 Jul 2023

Dear Dr Zeden,

We are delighted to inform you that your manuscript, "Metabolic reprogramming and altered cell envelope characteristics in a pentose phosphate pathway mutant increases MRSA resistance to β-lactam antibiotics," has been formally accepted for publication in PLOS Pathogens.

Best regards,

Kasturi Haldar

Editor-in-Chief

PLOS Pathogens

orcid.org/0000-0001-5065-158X

Michael Malim

Editor-in-Chief

PLOS Pathogens

orcid.org/0000-0002-7699-2064